# FBXL4 ubiquitin ligase deficiency promotes mitophagy by elevating NIX levels

Hannah Elcocks[1,†] ⓘ, Ailbhe J Brazel[1,†,‡] ⓘ, Katy R McCarron[1] ⓘ, Manuel Kaulich[2,3] ⓘ, Koraljka Husnjak[2] ⓘ, Heather Mortiboys[4] ⓘ, Michael J Clague[1,*] ⓘ & Sylvie Urbé[1,**] ⓘ

## Abstract

Selective autophagy of mitochondria, mitophagy, is linked to mitochondrial quality control and as such is critical to a healthy organism. We have used a CRISPR/Cas9 approach to screen human E3 ubiquitin ligases for influence on mitophagy under both basal cell culture conditions and upon acute mitochondrial depolarization. We identify two cullin-RING ligase substrate receptors, VHL and FBXL4, as the most profound negative regulators of basal mitophagy. We show that these converge, albeit via different mechanisms, on control of the mitophagy adaptors BNIP3 and BNIP3L/NIX. FBXL4 restricts NIX and BNIP3 levels via direct interaction and protein destabilization, while VHL acts through suppression of HIF1α-mediated transcription of BNIP3 and NIX. Depletion of NIX but not BNIP3 is sufficient to restore mitophagy levels. Our study contributes to an understanding of the aetiology of early-onset mitochondrial encephalomyopathy that is supported by analysis of a disease-associated mutation. We further show that the compound MLN4924, which globally interferes with cullin-RING ligase activity, is a strong inducer of mitophagy, thus providing a research tool in this context and a candidate therapeutic agent for conditions linked to mitochondrial dysfunction.

**Keywords** BNIP3; FBXL4; mitophagy; NIX; VHL

**Subject Categories** Autophagy & Cell Death; Organelles; Post-translational Modifications & Proteolysis

The EMBO Journal (2023) 42: e112799

See also: **LP Wilhelm & IG Ganley** (July 2023)

## Introduction

The selective disposal of damaged mitochondria by autophagy, also known as mitophagy, is a critical element of mitochondrial quality control. Defects in mitophagy have been linked to neurodegenerative ailments such as Parkinson's and Alzheimer's disease and many other pathophysiological conditions (Sorrentino *et al*, 2017; Rodolfo *et al*, 2018; Fritsch *et al*, 2019). In Parkinson's disease, some patients show loss of function in PINK1 or PRKN genes, which renders cells unable to clear damaged mitochondria (Agarwal & Muqit, 2022). *PRKN* encodes an RBR E3 ligase called Parkin, which is itself activated by PINK1 following damage to mitochondria. Parkin and ubiquitin are both direct substrates of PINK1, and pSer65-ubiquitin (pUb) is required for recruitment and full activation of Parkin, leading to a rapid feed-forward amplification loop and extensive ubiquitylation of mitochondrial outer membrane proteins (Bingol & Sheng, 2016; Harper *et al*, 2018; Pickles *et al*, 2018). This ubiquitin coat serves to recruit ubiquitin-binding domain-encoding adaptors (ubiquitin receptors) that link the mitochondrial and autophagosomal membranes, via LC3-interacting regions (LIR) (Lazarou *et al*, 2015).

Based on our past studies of PINK1- and Parkin-dependent mitophagy, we have proposed that other ubiquitin E3 ligases can prime a seed population of mitochondrial proteins with ubiquitin, which can then be phosphorylated by PINK1 to unleash the Parkin-mediated ubiquitylation cascade (Marcassa *et al*, 2018; Rusilowicz-Jones *et al*, 2020). In cell lines that do not express Parkin, we have observed a PINK1-dependent component of mitophagy that remains sensitive to depletion or deletion of the mitochondrial deubiquitylase USP30 (Marcassa *et al*, 2018). However, in organisms such as mice and flies, the majority of constitutive or basal mitophagy is independent of PINK1 and Parkin (Lee *et al*, 2018; McWilliams *et al*, 2018). Here, we have employed a CRISPR/Cas9 screening approach to identify further ubiquitin E3 ligases, which may regulate mitophagy.

All previous mitophagy screens have used acute mitochondrial depolarization as a trigger for mitophagy in cell systems overexpressing Parkin (Heo *et al*, 2019; Hoshino *et al*, 2019). Here, we have undertaken the technically more challenging task of screening for regulators of mitophagy in the absence of Parkin both with and without a depolarizing insult. We identify two cullin-RING E3 ligase

---

1 Molecular Physiology and Cell Signalling, Institute of Systems, Molecular and Integrative Biology, University of Liverpool, Liverpool, UK
2 Institute of Biochemistry II, Goethe University, Medical Faculty, University Hospital, Frankfurt am Main, Germany
3 Frankfurt Cancer Institute, Frankfurt am Main, Germany
4 Sheffield Institute for Translational Neuroscience (SITraN), University of Sheffield, Sheffield, UK
   *Corresponding author. Tel: +44 151 7945308; E-mail: clague@liv.ac.uk
   **Corresponding author. Tel: +44 151 7945432; E-mail: urbe@liv.ac.uk
   †These authors contributed equally to this work
   ‡Present address: Department of Biology, Maynooth University, Maynooth, Ireland

substrate adapters, VHL and FBXL4, as the most profound negative regulators under both basal and depolarizing conditions. VHL is a major tumour suppressor while FBXL4 has been linked to early-onset mitochondrial encephalomyopathy, also referred to as mitochondrial DNA depletion syndrome 13 (Bonnen et al, 2013; Gai et al, 2013; Gossage et al, 2015). FBXL4 is a substrate receptor of a Cullin-1-RING E3 (CRL1) ligase complex and has previously been shown to bind to SKP1, the common CRL1 adapter subunit (Winston et al, 1999; Tan et al, 2013). It encodes a mitochondrial targeting sequence and has been shown to associate with mitochondria (Bonnen et al, 2013; Gai et al, 2013). A recent study found increased levels of mitophagy in both FBXL4 KO mice and patient-derived fibroblasts, although the mechanism of action remained to be elucidated (Alsina et al, 2020).

The connection between VHL and mitophagy is well established, operating through the control of HIF-1/2 α stability and consequent effects upon transcription of the mitophagy adaptors BNIP3 and BNIP3L/NIX, hereafter referred to as NIX (Bellot et al, 2009). In distinction to mitophagy adaptors utilized by the PINK1–Parkin pathway, BNIP3 and NIX integrate into the outer mitochondrial membrane and then recruit the phagophore membrane via their LIR motifs, in common with other selective autophagy adaptors (Novak et al, 2010; Hanna et al, 2012). We now show that FBXL4 is a suppressor of BNIP3 and NIX, but in contrast to VHL, exerts its control through the alternative mechanism of direct interaction and protein destabilization. Analysis of an FBXL4 mutant, associated with mitochondrial DNA depletion syndrome 13, highlights the pathophysiological relevance of NIX as a substrate for FBXL4 and suggests a molecular explanation for the aetiology of the disease. Our study indicates that NIX can be considered the master regulator of basal mitophagy and we have now unveiled a novel mechanism through which it is controlled.

# Results

### CRISPR/Cas9 E3 ligase screen for mitophagy

We chose to utilize hTERT-RPE1 cells in our search for alternative ubiquitin E3 ligases that modulate mitophagy in the absence of Parkin. We have previously established that these cells respond to depletion and deletion of the deubiquitylase USP30 by an increase in basal mitophagy, indicating that alternative ubiquitin-dependent mitophagy pathways are in operation (Marcassa et al, 2018). We first introduced the mitochondrial matrix-targeted pH-sensitive mitophagy reporter mt-mKeima into a puromycin-sensitive clone of hTERT-RPE1 cells expressing inducible Cas9 (Cas9i). This fluorophore responds to the acidic pH of the lysosomal compartment by a shift in its excitation spectrum. Hence, it reports on the delivery of mitochondrial fragments to mature autophagosomes (mitolysosomes) and provides a ratiometric readout that can be monitored by flow cytometry (Katayama et al, 2011). We established that our cells exhibit a measurable response to mitochondrial depolarization using a 24 h treatment of respiratory chain inhibitors, antimycin and oligomycin (AO). This resulted in a > 2-fold increase in the population of cells captured in the high mitophagy gate (Fig EV1A–E).

To initiate the fluorescence-based CRISPR KO screen, RPE1-Cas9i-mt-mKeima cells were transduced with a lentiviral E3 ligase sgRNA library, targeting 606 Ubiquitin E3 ligases, at a multiplicity of infection (MOI) of 0.2 and a 1,000× library coverage. Cells were then grown in puromycin-containing media to select for integration of sgRNAs prior to induction with doxycycline to initiate Cas9 expression and gene targeting (Fig 1A). Half of the cells were treated with AO for 24 h to induce mitophagy and then sorted alongside untreated cells (basal mitophagy) for both high (enhanced) and low (decreased) mitophagy. An unsorted sample of each population was collected for reference. Genomic DNA was harvested, barcoded sgRNA libraries amplified and two independent replicates (pooled 2 by 2 from four independent biological experiments) were analysed by next-generation sequencing (NGS). Enriched sgRNAs were identified using the Screen Processing Pipeline (Horlbeck et al, 2016), obtaining for each comparison both a gene-level enrichment value as well as a P-value comparing the combined enrichment of all four sgRNAs targeting an individual E3 to that of 243 non-targeting control (NTC) sgRNA values in the same replicate (Figs 1B and C, and EV1F and G, average values shown; Dataset EV1). In contrast to other mitophagy screens performed in Parkin-overexpressing cells (Heo et al, 2019; Hoshino et al, 2019), we anticipated a narrow dynamic range, especially in our experiment designed to monitor basal mitophagy.

Any sgRNAs that knock out a positive regulator of mitophagy are expected to be enriched in the "Low" (decreased) mitophagy gates and de-enriched in the "High" (enhanced) mitophagy gates. We assembled an initial candidate list of E3 ligases that met (de)-enrichment criteria ($P < 0.05$; $-0.2 > $ Log$_2$ fold change $> 0.2$) in at least one comparison in either replicate. A handful of E3s returned $P$-values $< 0.05$ in two replicates either in the "High" or "Low" gates in basal or induced mitophagy (Fig 1D and E). Under mitochondrial depolarizing conditions, our screen presumably represents the sum of basal and AO-induced mitophagy and it is thus expected that critical regulators of basal mitophagy are identified in both screens. Indeed, we identify nine E3s that are common to both screens and only four that are unique to basal mitophagy (Fig 1F). Common positive regulators of basal and AO-induced mitophagy include two subunits of the homotypic fusion and vacuolar protein sorting (HOPS) complex (VPS41 and VPS18; VPS11 scores do not pass the thresholds set here). This provided us with confidence that our screen was identifying meaningful hits as these, and/or other HOPS complex subunits, have previously been identified in mitophagy and global autophagy screens (Moretti et al, 2018; Morita et al, 2018; Heo et al, 2019; Hoshino et al, 2019; Jia & Bonifacino, 2019; Shoemaker et al, 2019) and are well understood to regulate membrane fusion of autophagosomes and endosomes with lysosomes (Jiang et al, 2014; Takats et al, 2014). Five positive and two negative regulators are only detected in the AO-induced mitophagy screen (Fig 1F).

### VHL and FBXL4 repress BNIP3 and NIX protein levels by different mechanisms

Here, we have focused our attention on the two major outliers of the basal mitophagy screen, VHL and FBXL4, for which the knock-out results in a strong increase in mitophagy (Fig 1B–F). VHL acts as the substrate recognition subunit of a Cullin-2-RING E3 ligase complex and is responsible for the constitutive turnover of the transcription factors HIF1α and HIF2α. Hypoxic conditions result in

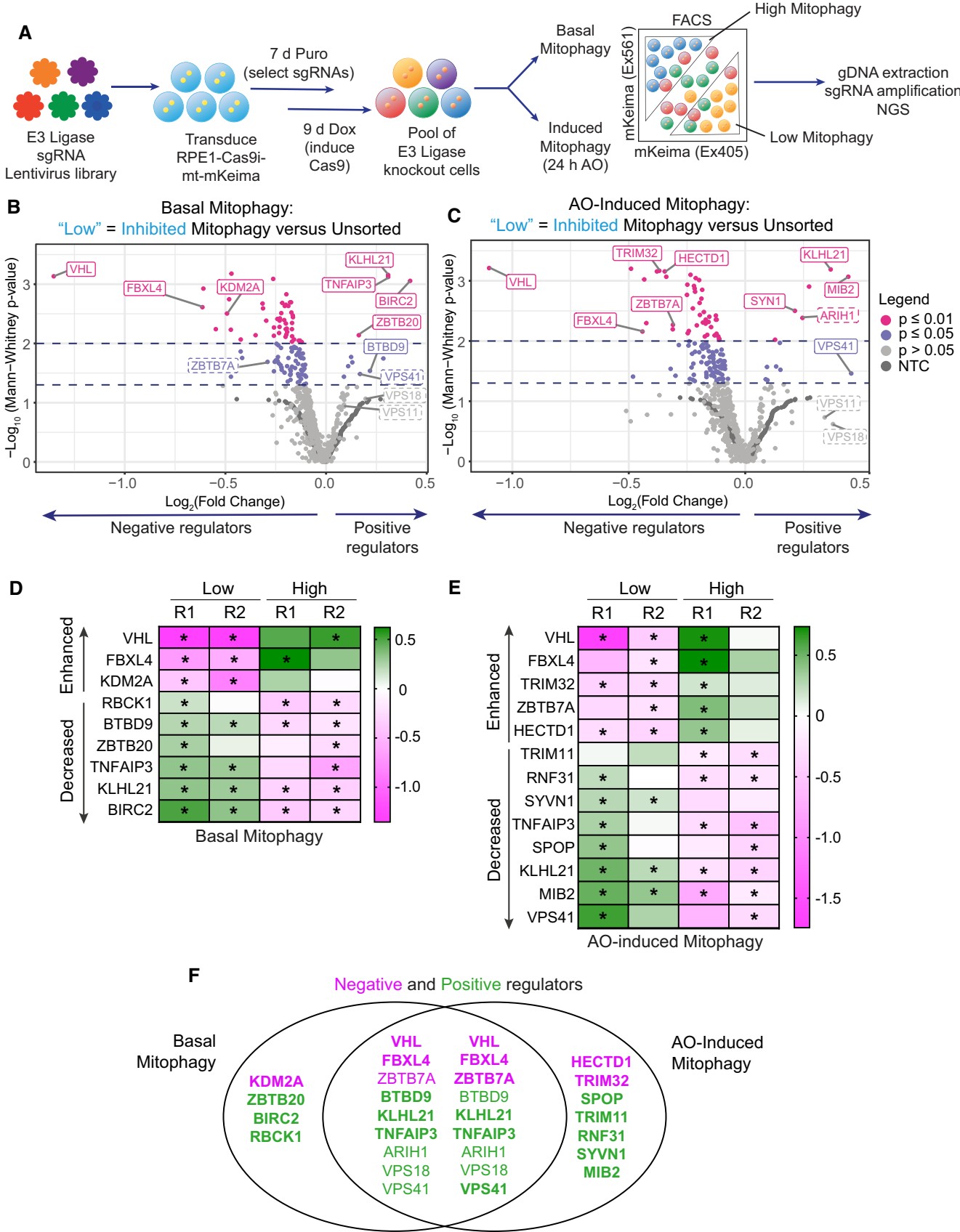

**Figure 1.**

**Figure 1.  CRISPR screen targeting E3 ligases identifies major regulators of Parkin-independent mitophagy.**

A  Schematic depicting the CRISPR screening strategy. RPE1-Cas9i-mt-mKeima cells were transduced with a lentiviral CRISPR sgRNA library targeting 606 E3 ligases. sgRNA-expressing cells were selected for 7 days with puromycin (Puro) and Cas9 expression induced with doxycycline (Dox) for 9 days. Fourteen days post-transduction, half the cells were treated with antimycin and oligomycin (AO; 1 and 10 μM respectively) for 24 h, then sorted alongside untreated cells (basal mito-phagy) by FACS into "high" and "low mitophagy" populations based on mt-mKeima fluorescence. Genomic DNA was extracted from sorted and unsorted reference cells, sgRNAs amplified by barcoded PCRs and samples analysed by next-generation sequencing (NGS).

B, C  Volcano plot showing the average $\log_2$ fold change and $-\text{Log}_{10}$ $P$-value of genes in low mitophagy versus unsorted cells in the AO-induced and basal mitophagy screen for two independent biological replicates. Statistical thresholds of 2 and 3 standard deviations from the mean are indicated by dashed lines and colour coding. Indicated are high-confidence (unbroken line) and lower-confidence (dashed line) candidates shown in (D–F).

D, E  High-confidence candidate list of positive (decreased) and negative (enhanced) regulators of Parkin-independent-induced mitophagy. Heatmap showing the $\text{Log}_2$ fold change of genes in low/high mitophagy versus unsorted cells in the induced and basal mitophagy screen for each of two independent biological replicates. Genes with $P$-values < 0.05 are indicated by an asterisk.

F  Venn diagram showing the overlap of genes listed in (D) and (E) that were identified in the basal and AO-induced mitophagy screens as positive (green) and negative (magenta) regulators. Bold type indicates high-confidence hits and regular type indicates lower-confidence hits.

Source data are available online for this figure.

disengagement of HIF1α and HIF2α from VHL leading to their stabilization and the transcription of a large number of genes, including the paralogues BNIP3 and NIX (Bellot *et al*, 2009). We wondered if FBXL4 loss may likewise promote mitophagy by upregulating BNIP3 and/or NIX.

We first adopted an orthogonal approach of depleting VHL and FBXL4 in a parental hTERT-RPE1 FlpIN cell line with pools of siRNAs. As none of the available antibodies against FBXL4 was able to detect endogenous protein levels, we used qRT–PCR to confirm its depletion (Fig EV2A). We found a clear increase in both NIX and to a lesser extent BNIP3 protein levels in FBXL4-depleted cells, whereas BNIP3 rather than NIX was upregulated in response to VHL depletion (Fig 2A and B). Both BNIP3 and NIX basal protein levels were sensitive to HIF1α depletion. Importantly, in contrast to VHL knockdown, depletion of FBXL4 did not stabilize HIF1α, providing a first indication for a distinct mechanism controlling BNIP3 and NIX protein levels. Evidence that this is non-transcriptional is provided by the observation that neither BNIP3 nor NIX transcript levels

increase in FBXL4-depleted cells, while both transcripts were clearly sensitive to HIF1α depletion (Fig EV2B).

We next chose two sgRNAs from the E3 library mitophagy screen to generate a pair of knockout (KO) cell pools. We transduced the RPE1-Cas9i-mt-mKeima cells with the respective sgRNA-encoding lentivirus, selected for stable sgRNA integration using puromycin selection and then induced Cas9 to initiate the knockout, all the while maintaining a non-induced pool of cells as a control population. While keeping these KO cells as pools means that a fraction of the alleles are non-edited (e.g. ~30% wild-type for FBXL4, Fig EV2C), this has the advantage of avoiding the issues associated with clonal variability. Both KO pools, generated with independent sgRNAs, showed a clear increase in both BNIP3 (3–4×) and NIX (5–7×) protein levels and a significant increase in basal mitophagy that can be measured by both flow cytometry and live cell imaging (Fig 3A–F). A recent proteomic analysis comparing FBXL4 KO mice and patient-derived fibroblasts reported decreased levels of a large number of mitochondrial proteins with a concomitant increase in

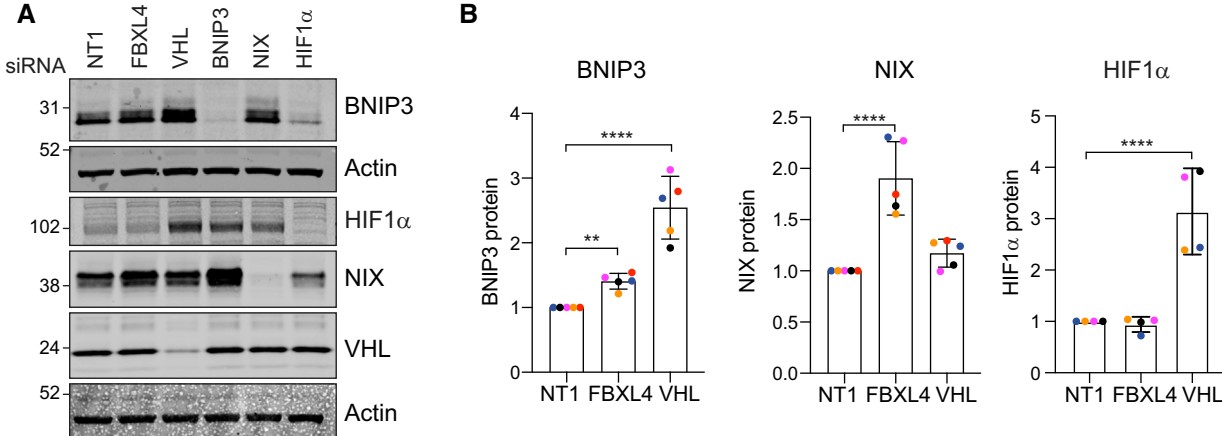

**Figure 2.  VHL and FBXL4 control expression of mitophagy adaptors BNIP3 and NIX.**

A  Representative western blot of hTERT-RPE1 cells transfected for 72 h with siRNA against indicated targets or non-targeting (NT1) prior to harvesting and cell lysis.

B  Quantification of BNIP3, NIX and HIF1α protein levels for data represented in A. Individual data points from $n = 5$ (BNIP3, NIX) or 4 (HIF1α) independent colour-coded experiments are shown. Shown is the fold change normalized to NT1 control. Error bars show standard deviation. One-way ANOVA and Dunnett's multiple-comparison test, **$P$ < 0.01 and ****$P$ < 0.0001.

Source data are available online for this figure.

lysosomal proteins (Alsina *et al*, 2020). Our FBXL4 KO RPE1 cells also displayed a moderate loss of the outer mitochondrial protein TOMM40, but we did not observe any significant changes in cathepsin D, one of the lysosomal enzymes highlighted in the preceding study (Fig EV2D–F). As expected, immunofluorescence revealed a mitochondrial localization for this excess BNIP3 and NIX (Fig 3C). Importantly, we did not observe any changes in NIX or BNIP3 transcript levels, or in HIF1α protein levels, further indicating a post-transcriptional regulatory mechanism at play (Fig 3G and H).

## FBXL4 interacts with BNIP3 and NIX and governs their stability

We wondered whether BNIP3 and NIX could be physiologically relevant substrates of the CRL$^{FBXL4}$. Consistent with this hypothesis, reciprocal pulldown experiments using GFP-tagged BNIP3 or NIX and Flag-tagged FBXL4 confirmed their interaction (Fig 4A and B).

Disease-associated mutations in FBXL4 are largely focused on the leucine-rich region (LRR), which in other FBXL family proteins mediates substrate recruitment (Bonnen *et al*, 2013; Gai *et al*, 2013; Ballout

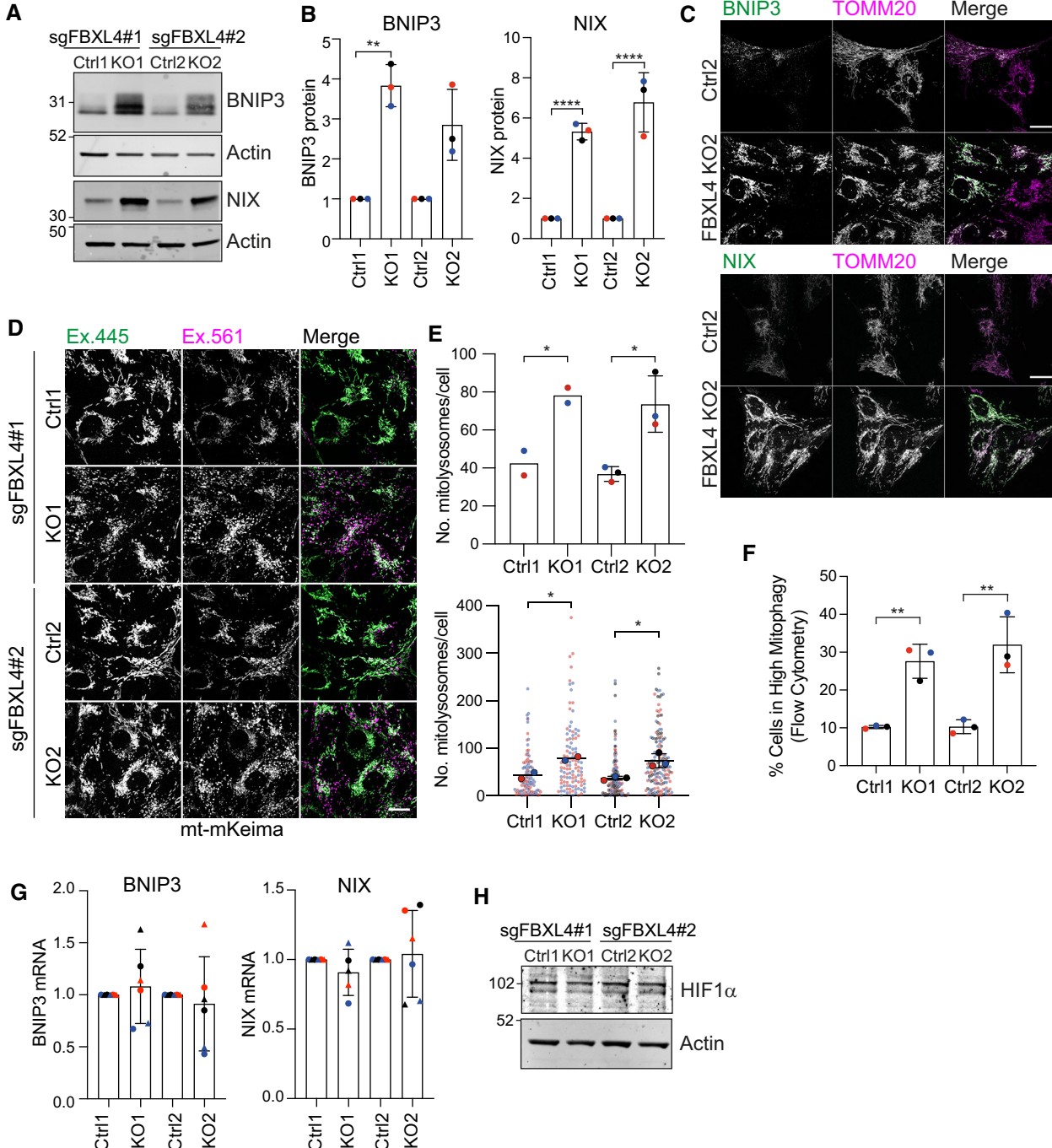

**Figure 3.**

◀

**Figure 3.  FBXL4 knockout upregulates BNIP3 and NIX and enhances basal mitophagy.**

A    Representative western blot of two matched pairs of control (Ctrl) and FBXL4 KO pool RPE1-Cas9i-mt-mKeima cells, generated with two distinct sgRNAs.

B    Quantification of data represented in A. Individual data points from $n = 3$ independent colour-coded experiments are shown. Error bars show standard deviation of the fold change in BNIP3 and NIX in each KO pool normalized to their matched control pools; one-way ANOVA and Tukey's multiple-comparison test, $**P < 0.01$ and $****P < 0.0001$.

C    Representative images of Ctrl and FBXL4 KO pool RPE1-Cas9i-mt-mKeima cells fixed and stained for either BNIP3 or NIX and co-stained for TOMM20. Scale bar 20 μm.

D    Representative images of control or FBXL4 KO pool RPE1-Cas9i-mt-mKeima cells. "Magenta only" puncta (Ex. 561) correspond to acidic mitolysosomes. Scale bar 20 μm.

E    Graphs show the quantification of mitolysosomes (mt-mKeima "magenta only" puncta) per cell. A minimum of 45 cells were analysed per condition in each experiment. Individual data from two (for control 1 (Ctrl 1) and knockout 1 (KO1)) or three (control 2 (Ctrl 2) and knockout 2 (KO2)) are shown using different colours for each independent experiment. Mean and standard deviation are indicated. Bottom graph shows the individual data points. One-way ANOVA with Tukey's multiple-comparison test, $*P < 0.05$.

F    Basal mitophagy was analysed in Ctrl and FBXL4 KO RPE1-Cas9i-mt-mKeima cells by flow cytometry. High mitophagy gates were set using the corresponding control for each knockout. Three independent colour-coded experiments are shown; one-way ANOVA and Tukey's multiple-comparison test, $**P < 0.01$.

G    Quantitative RT–PCR reactions of BNIP3 and NIX (normalized to actin) were performed with cDNA derived from control and FBXL4 KO RPE1-Cas9i-mt-mKeima cells. Two primer pairs were used per target and are indicated by symbol shape. Three independent colour-coded experiments are shown. Error bars show standard deviation of the fold change of BNIP3 and NIX mRNA in each KO pool normalized to their matched control pools.

H    Representative western blot of control and FBXL4 KO RPE1-Cas9i-mt-mKeima cells probed for HIF1α.

Source data are available online for this figure.

*et al*, 2019). We selected a single LRR-domain point mutant R482W and assessed its ability to interact with BNIP3 and NIX. Surprisingly, both NIX and BNIP3 were still able to co-immunoprecipitate with this disease-associated FBXL4 mutant protein while its interaction with SKP1 was reduced by half (Fig 4C–E). We also note that FBXL4-Flag presents with an as-yet-uncharacterized higher-molecular-weight band that is not enriched in the GFP-BNIP3 or -NIX pulldowns (Fig 4B and C, blue arrowheads).

We next asked whether either wild-type or mutant FBXL4 would be able to rescue NIX and BNIP3 protein levels in FBXL4 KO cells. Immunofluorescence microscopy of endogenous BNIP3 and NIX showed that overexpression of wild-type FBXL4-Flag was able to suppress both BNIP3 and NIX, whereas the FBXL4 LRR mutant [R482W] was not (Fig 4F and G). Although only a fraction of cells expressed the transgenes, a clear decrease in NIX and BNIP3 protein levels was also seen by immunoblotting lysates from cells transfected with wild-type but not mutant FBXL4 (Fig 4H).

### NIX is the principal regulator of mitophagy

We reasoned that if the increased levels of BNIP3 and/or NIX are the root cause for the enhanced mitophagy in FBXL4 KO cells, then we should be able to restore low mitophagy levels by depleting these factors. We transfected control and FBXL4 KO cells with siRNA against BNIP3, NIX and HIF1α and assessed basal mitophagy levels. We visualized mitophagy using fluorescence imaging of the mt-mKeima reporter (Fig 5A). In addition, we used FACS analysis as a highly quantitative readout for basal mitophagy levels and verified knockdown efficiency using western blotting (Fig 5B and C). Both approaches show that depletion of NIX, but neither BNIP3 nor HIF1α, reduces basal mitophagy levels to that of the control cells. These results identify NIX rather than BNIP3 as the primary mediator of enhanced mitophagy observed in FBXL4 KO cells.

### Enhancement of mitophagy by MLN4924

Both VHL and FBXL4 are substrate recognition modules of CRL E3 ligase complexes, which require neddylation for their activation. Global inactivation of CRLs can be acutely achieved with MLN4924, a highly selective compound that inhibits the Nedd8-E1-conjugating enzyme (Brownell *et al*, 2010). We surmised that MLN4924 application may provide a potent stimulus to mitophagy through upregulation of BNIP3 and NIX.

▶

**Figure 4.  FBXL4 interacts with and destabilizes BNIP3 and NIX.**

A, B    hTERT-RPE1 cells were co-transfected with Flag (V) or FBXL4-Flag (FL4) and GFP (Vect), GFP-BNIP3 or GFP-NIX. Lysates were subjected to immunoprecipitation (IP) with either GFP-nanobody-coupled beads (A) or Flag-antibody-coupled agarose beads (B). IPs were probed alongside the input samples for Flag and GFP.

C    hTERT-RPE1 cells were co-transfected with FBXL4-Flag (WT) or FBXL4-R482W-Flag (RW) and GFP, GFP-BNIP3 (BNIP3) or GFP-NIX (NIX). Lysates were subjected to immunoprecipitation with GFP-nanobody-coupled beads, and IPs were probed alongside the input samples for Flag and GFP.

D    hTERT-RPE1 cells were transfected with empty vector (V), FBXL4-Flag (WT) or FBXL4-R482W-Flag (RW). Lysates were subjected to IP with Flag-antibody-coupled agarose beads and probed for SKP1 and Flag. Note that FBXL4-Flag presents with an as-yet-uncharacterized higher-molecular-weight band (indicated by a blue arrowhead) that is not enriched in the GFP-BNIP3 or -NIX pulldowns (A–D).

E    Quantitation of SKP1 protein co-immunoprecipitated with FBXL4-R482W-Flag relative to FBXL4-Flag (WT). Data from two independent experiments are shown, normalized to the amount of SKP1 in the input and FBXL4-Flag in the IP.

F, G    Representative images of control (Ctrl) and FBXL4 KO RPE1-Cas9i-mt-mKeima cells transfected with FBXL4-Flag wild-type (WT) or FBXL4-Flag [R482W] for 24 h, then fixed and stained for either BNIP3 or NIX. Scale bar 20 μm. Quantitation of the frequency of high BNIP3 and high NIX phenotypes observed in (F) and (G). A minimum of 32 Flag-positive (transfected, indicated by an asterisk in the images) and Flag-negative cells were analysed per condition in each experiment. Shown are the mean and individual data points of two colour-coded experiments.

H    Western blot of control and FBXL4 KO RPE1-Cas9i-mt-mKeima cells transfected as in (F) and (G) with Flag (V), FBXL4-Flag (WT) or FBXL4-Flag [R482W] (RW) for 24 h.

Source data are available online for this figure.

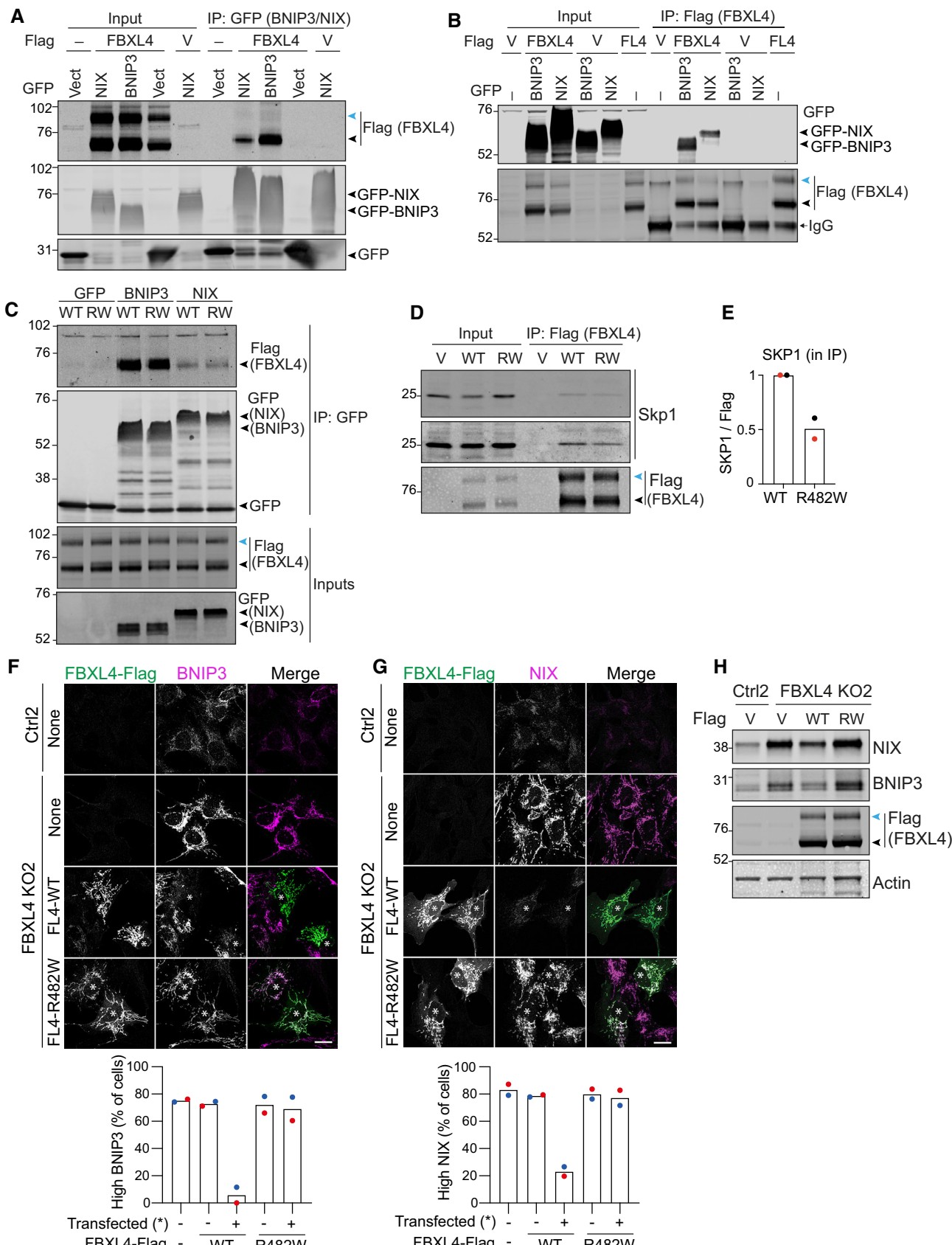

Figure 4.

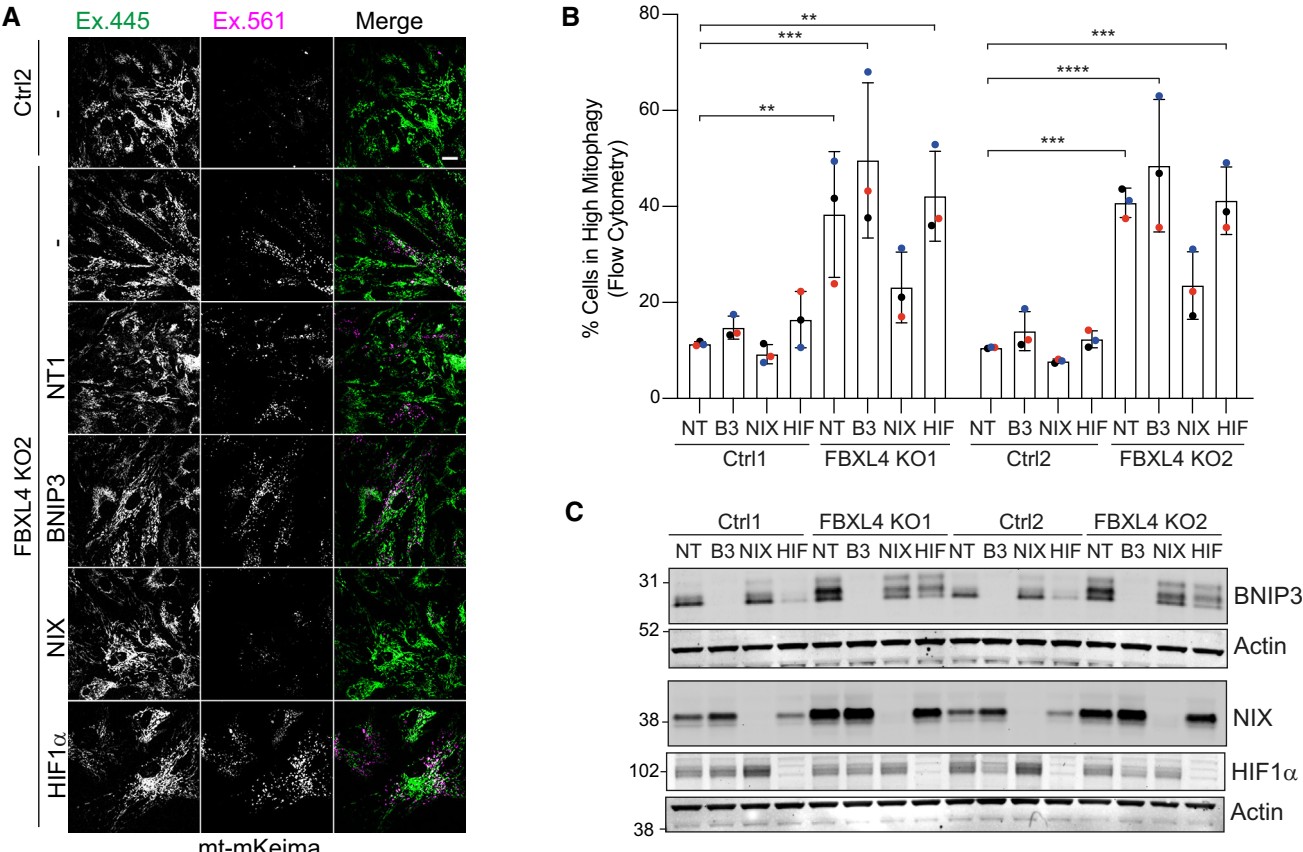

**Figure 5. Enhancement of basal mitophagy upon FBXL4 deletion is NIX dependent.**

A, B   Control (Ctrl) or FBXL4 KO RPE1-Cas9i-mt-mkeima cells were treated for 72 h with non-targeting (NT1), BNIP3 (B3), NIX or HIF1α (HIF) siRNA. (A) Representative images of the mt-mKeima reporter in control (Ctrl2) or FBXL4 KO2 RPE1-Cas9i-mt-mkeima. "Magenta only" puncta (Ex. 561) correspond to acidic mitolysosomes. Scale bar: 20 μm. (B) Flow cytometry analysis providing a quantitative assessment of basal mitophagy. High mitophagy gates were set for each matched pair of control and KO cells using the non-targeting (NT) siRNA-treated control pool sample as a reference. Data shown represent three colour-coded independent experiments. Error bars show standard deviation. One-way ANOVA and Dunnett's multiple-comparison test, **$P < 0.01$, ***$P < 0.001$ and ****$P < 0.0001$.

C      Representative western blot verifying the knockdown efficiency for samples shown in (B).

Source data are available online for this figure.

We first treated mt-mKeima-expressing hTERT-RPE1 cells for 24 h with MLN4924 and assessed the impact on basal mitophagy by fluorescence microscopy. We observed a dramatic increase in mitophagy, which surpasses the levels elicited with a depolarizing trigger (AO) over the same time period (Fig 6A). Both BNIP3 and NIX protein levels are strongly increased in the MLN4924 but not AO-treated samples, whereas only AO treatment resulted in PINK1 stabilization and an associated pSer65-ubiquitin signal (Figs 6B–D and EV3A and B). The increase of BNIP3 and NIX proteins is seen as early as 6 h after application of MLN4924 (Fig 6E). Both BNIP3 and NIX protein increase is mirrored by an increase in transcripts (Fig EV3C) but only BNIP3, not NIX, protein levels are sensitive to HIF1α depletion (Fig 6F and G). The MLN4924-induced increase in basal mitophagy is sensitive to NIX- but not BNIP3 depletion alone, although a double knockdown is required for full restoration (Fig 6H). Depletion of HIF1α only has a marginal effect on MLN4924-induced mitophagy, which provides further support for NIX as the main mediator of this pathway (Fig 6I).

By 24 h of MLN4924 treatment, BNIP3 protein levels increase by > 20× as compared to just 3–4× seen in the FBXL4 KO cells, whereas the fold increase for NIX is comparable (7×) (Figs 3B and 6D). Importantly, in FBXL4 KO cells, NIX but not BNIP3 protein levels were refractory to MLN4924, providing strong evidence for FBXL4 as the key neddylation-dependent mediator of NIX (Fig 7A).

We noticed that the MLN4924-induced NIX protein had a shorter half-life than the basally expressed pool (Fig EV3D), providing an opportunity to explore the impact of FBXL4 on NIX and BNIP3 half-lives. We pre-treated our isogenic cell pairs (±FBXL4) for 6 h with MLN4924 to accumulate NIX and BNIP3 and then following MLN4924 washout, performed a cycloheximide chase experiment. NIX and BNIP3 are stabilized in the absence of FBXL4 and the turnover of this labile pool can be rescued by inhibiting the proteasome (Figs 7B–E and EV3E–G). Finally, we show that the ubiquitylated fraction of NIX is markedly reduced in the FBXL4 KO cells (Fig 7F and G; note some residual ubiquitylation is expected from ~30% wild-type alleles in the cell pool).

# Discussion

The critical role of mitophagy in multiple physiological settings has been established (Schweers *et al*, 2007; Esteban-Martinez & Boya, 2018; Harper *et al*, 2018; Zhao *et al*, 2020; Onishi *et al*, 2021; Ordureau *et al*, 2021; Teresak *et al*, 2022). Given the variety of

contexts in which it occurs, it is not surprising that multiple mechanisms of induction and control exist. Previous mitophagy CRISPR screens have focused on the PINK1-PRKN–dependent pathway, which is elicited by acute mitochondrial depolarization (Heo *et al*, 2019; Hoshino *et al*, 2019). Here, we have diverged in using a cell line which does not express Parkin and have given equal emphasis

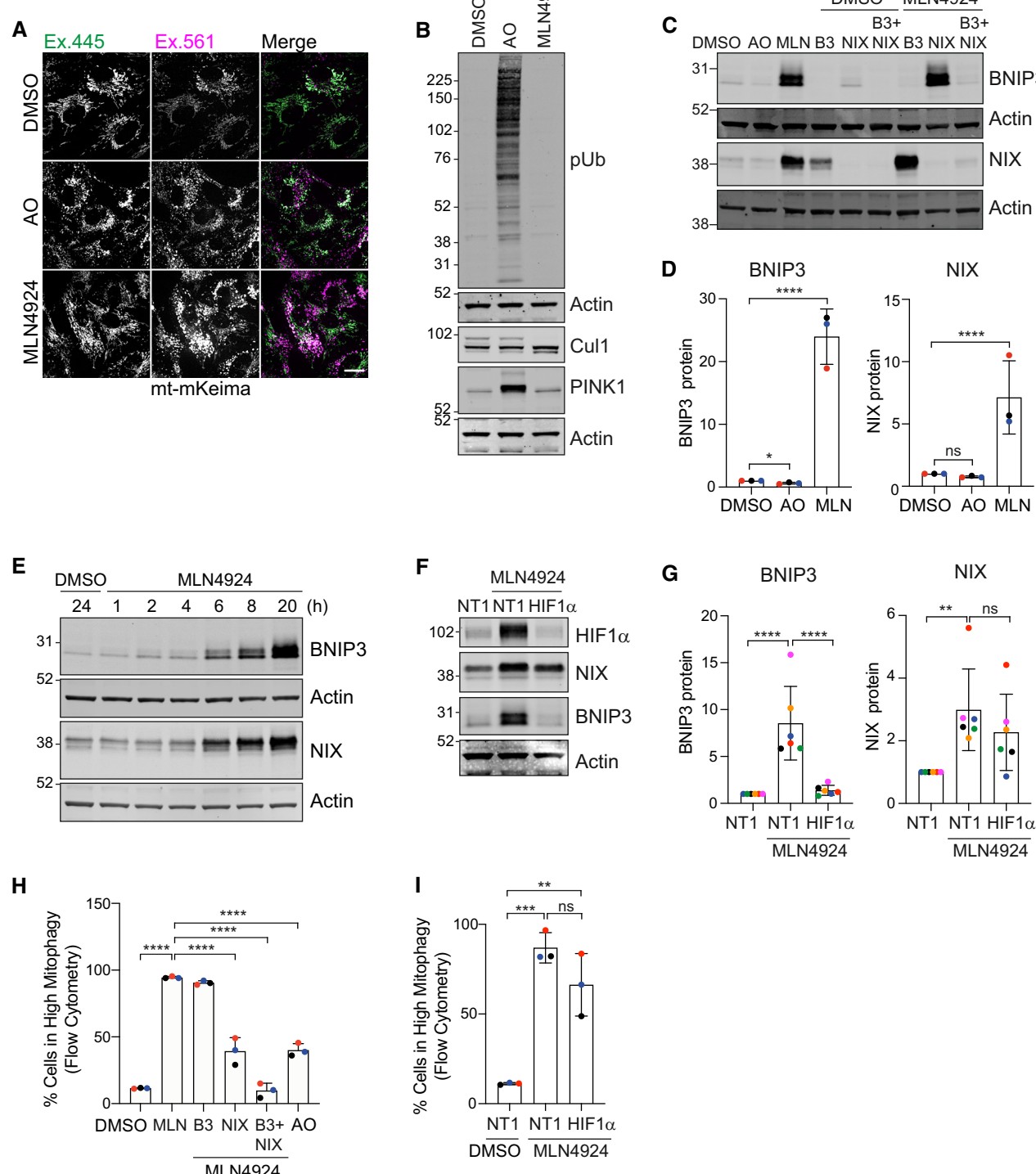

**Figure 6.**

**Figure 6.  The neddylation inhibitor MLN4924 upregulates BNIP3 and NIX and enhances mitophagy.**

A  Representative images of RPE1-Cas9i-mt-mKeima cells treated with antimycin and oligomycin (AO; 1 and 10 μM, respectively) or MLN4924 (1 μM) for 24 h. Scale bar 20 μm.

B  Western blot analysis of lysates prepared from hTERT-RPE1 cells treated with antimycin and oligomycin (AO; 1 and 10 μM, respectively) or MLN4924 (1 μM) for 24 h.

C  Representative western blot of lysates prepared from hTERT-RPE1 cells transfected with siRNA-targeting BNIP3 (B3), NIX or both for 72 h. Cells were treated with either DMSO or MLN4924 (1 μM) for the last 24 h of transfection. Untransfected cells were treated alongside transfected ones with DMSO, antimycin and oligomycin (AO; 1 and 10 μM, respectively), or MLN4924.

D  Quantification of fold change in protein levels calculated from data represented in (C) for three colour-coded independent experiments; one-way ANOVA and Dunnett's multiple-comparison test, *$P < 0.05$ and ****$P < 0.0001$.

E  Representative western blot of hTERT-RPE1 cells treated with either DMSO or MLN4924 (1 μM) for the indicated time points.

F  Representative western blot of hTERT-RPE1 cells transfected with non-targeting (NT1) of HIF1α-targeting siRNA. Cells were treated with MLN4924 (1 μM) for the last 24 h of transfection.

G  Quantification of fold change in protein levels calculated from data represented in (F) for six colour-coded independent experiments. Error bars show standard deviation; one-way ANOVA and Tukey's multiple-comparison test, **$P < 0.01$ and ****$P < 0.0001$.

H  Mitophagy was measured in RPE1-Cas9i-mt-mkeima cells treated as in (C) using flow cytometry. Cells were transfected with siRNA-targeting BNIP3 (B3), NIX or both (DKD) for 72 h. Cells were treated with either DMSO or MLN4924 (1 μM) for the last 24 h of transfection. Untransfected cells were treated alongside transfected ones with DMSO, antimycin and oligomycin (AO; 1 and 10 μM respectively), or MLN4924.

I  Mitophagy was measured in RPE1-Cas9i-mt-mkeima cells using flow cytometry. Cells were transfected with non-targeting (NT1)- or HIF1α (HIF)-targeting siRNA and were treated with DMSO or MLN4924 (1 μM) for the last 24 h of transfection.

Data information: Data shown in (H) and (I) represent three colour-coded independent experiments. Error bars show standard deviation; one-way ANOVA and either Tukey's (I) or Dunnett's (H) multiple-comparison test, **$P < 0.01$, ***$P < 0.001$ and ****$P < 0.0001$.

Source data are available online for this figure.

to an experimental condition that does not employ depolarization, namely basal mitophagy. This condition, which is PINK1-PRKN independent, likely corresponds to the majority of mitophagy in living organisms (Lee *et al*, 2018; McWilliams *et al*, 2018). Many of the top hits from our screen are shared between basal and depolarizing conditions, suggesting that the basal mitophagy mechanisms are still extant alongside new pathways that are induced by depolarization. However, we have also identified certain E3s, such as MIB2, that specifically promote mitophagy under depolarizing conditions. These will be pursued in future studies, not least for a role in priming the PINK1-PRKN cascade.

We here identified VHL as a strong suppressor of mitophagy under both basal and depolarizing conditions. The mechanism by which VHL controls NIX/BNIP3 expression, and hence mitophagy, via suppression of HIF1α is well understood. FBXL4 and KDM2A were identified as other strong suppressors of basal mitophagy. Interestingly, KDM2A is a lysine demethylase, which is upregulated by HIF1α (Batie *et al*, 2017). FBXL4 more closely corresponds with VHL, appearing as a strong suppressor hit in both conditions. We have now been able to unravel why this is so. Both VHL and FBXL4, but not KDM2A, control BNIP3 and NIX levels but by different mechanisms (Figs 7H and EV3H). The transcriptional control, exerted by VHL, is particularly deterministic for BNIP3 expression

levels, which are highly sensitive to HIF1α depletion. In contrast, FBXL4 does not impact either BNIP3 or NIX transcript levels but rather acts at a post-translational level by promoting ubiquitin- and proteasome-dependent turnover of both proteins.

In our system, NIX, rather than BNIP3, is the crucial regulator of mitophagy. FBXL4 interacts with and regulates the stability of NIX and BNIP3. A disease-associated LRR-domain mutant is unable to restore elevated levels of both proteins seen in FBXL4-KO cells, despite retaining the ability to interact with both proteins. The reduced association with SKP1 indicates a partial failure of this mutant to assemble into a functional CRL complex, albeit further structure–function analysis will be required to elucidate this point.

Proteins often display non-exponential decay rates reflective of multiple pools (McShane *et al*, 2016; Rusilowicz-Jones *et al*, 2022). Global studies of protein decay rates have revealed that supernumerary copies of proteins operating within complexes are frequently unstable. Here, we find that overproduction of NIX in response to treatment with the neddylation inhibitor MLN4924 leads to a short-lived pool that is turned over in an FBXL4-dependent fashion.

We have introduced MLN4924 as a new tool to promote basal mitophagy, predominantly through the combined inhibition of VHL and FBXL4. This will complement the widely used iron chelator deferiprone (DFP), which likely promotes mitophagy through

**Figure 7.  FBXL4 controls NIX and BNIP3 stability.**

A  Representative western blot of lysates from control or FBXL4 KO RPE1-Cas9i-mt-mKeima cells treated with MLN4924 (1 μM) for 6 h.

B  Representative western blot of lysates from control or FBXL4 KO RPE1-Cas9i-mt-mKeima cells pre-treated without or with MLN4924 (1 μM) for 6 h followed by a cycloheximide (CHX) chase (100 μg/ml).

C, D  Quantification of protein levels calculated from data represented in (B) for three colour-coded independent experiments. Error bars show standard deviation.

E  Western blot of lysates from control (Ctrl2) or FBXL4 KO2 RPE1-Cas9i-mt-mKeima cells pre-treated with MLN4924 for 6 h. Cells were then either lysed or MLN4924 was chased out with or without MG132 (10 μM) for 4 h before lysis.

F  TUBES pulldown of ubiquitylated NIX. FBXL4 KO RPE1-Cas9i-mt-mkeima cells and matched control cells were treated as in E and lysates were subjected to a TUBES pulldown.

G  Quantitation of the amount of ubiquitylated NIX (Ub-NIX) isolated in the TUBES-pulldown normalized to total NIX protein levels in the input samples as shown in F. Data shown are from three independent colour-coded experiments. Error bar shows standard deviation; unpaired *t*-test, ****$P < 0.0001$.

H  Schematic of proposed dual control of BNIP3 and NIX expression levels by VHL and FBXL4.

Source data are available online for this figure.

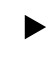

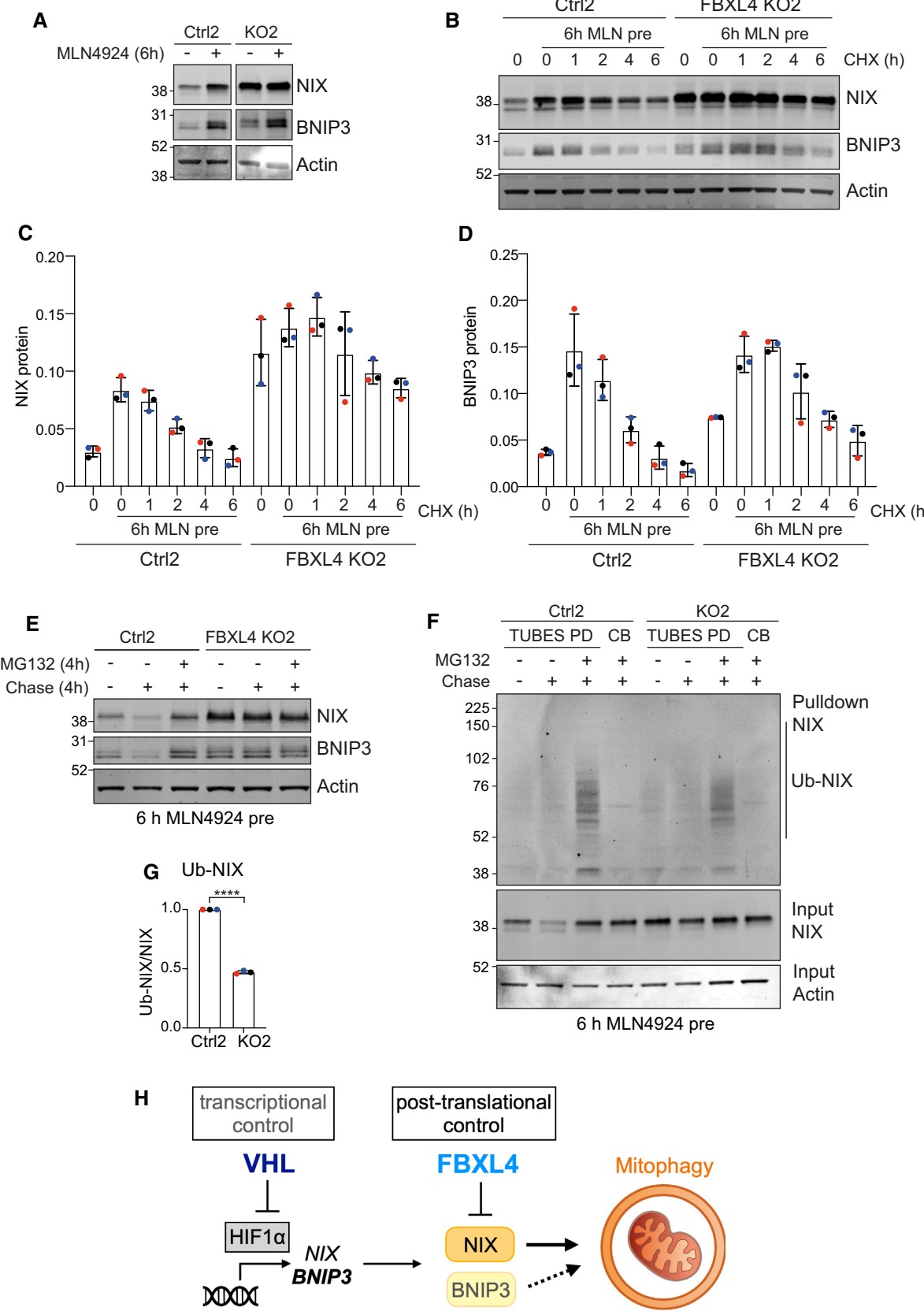

**Figure 7.**

inhibition of the iron-dependent enzyme, hypoxia-inducible factor prolyl hydroxylase (Allen *et al*, 2013; Zhao *et al*, 2020). As the clinical safety profile of MLN4924 is well characterized, it could be safely repurposed to stimulate mitophagy in patients who might have accumulated damaged mitochondria. Periodic stimulation of non-selective autophagy by TOR inhibitors has been shown to benefit mice-bearing neurodegenerative conditions associated with aggregate formation (Ravikumar *et al*, 2004; Rubinsztein, 2006; Menzies *et al*, 2010). As a corollary, one must also consider the impact of enhanced mitophagy in ongoing clinical trials for other indications. In this context, one interesting link between FBXL4 and VHL has been made in a genome-wide CRISPR screen for synthetic lethality with common tumour suppressors. Loss of FBXL4 is synthetically lethal with loss of VHL, which based on our findings, we speculate, is due to super-elevated levels of NIX and/or BNIP3 (Feng *et al*, 2022).

In conclusion, we have made a mechanistic link between FBXL4 and NIX in setting the levels of basal mitophagy that can add to the molecular understanding of the aetiology of early-onset mitochondrial encephalomyopathy. Our screening efforts highlighted two CRLs as critical regulators of mitophagy leading us to test MLN4924 as a pharmacological mitophagy-inducing agent. We expect this will take a prominent place in the mitophagy toolbox and look forward to exploring the clinical implications.

# Materials and Methods

### Generation of an RPE1-Cas9i-PuroS cell line

To generate a puromycin-sensitive version of the CT33-hTERT-RPE1-Cas9i cell line (received as a gift from Ian Cheeseman), the puromycin acetyltransferase (PAC) gene was targeted with CRISPR-Cas9. Cells were transiently transfected with pSpCas9(BB)-2A-GFP (PX458; 48138; Addgene) vector encoding a sgRNA-targeting PAC (5′-TGTCGAGCCCGACGCGCGTG-3′) using Lipofectamine 2000 (Invitrogen). GFP-positive cells were isolated by FACS 2 days after transfection. Single clones were isolated by screening for susceptibility to 3 μg/ml puromycin, and indel formation was confirmed by Sanger sequencing of a PCR product isolated from genomic DNA. The selected clone, hereafter referred to as RPE1-Cas9i, contained an 18 bp deletion in the PAC gene.

### Generation of the mitophagy reporter RPE1-Cas9i-mt-mKeima cell line

A mammalian expression construct for mKeima targeted to the mitochondrial matrix via a COXVIII-derived targeting sequence (AB01-pcDNA3.1-mt-mKeima-BlastR) was transiently transfected into RPE1-Cas9i cells using Lipofectamine 2000 (Invitrogen). Transfected cells were selected with 10 μg/ml blasticidin S HCl for 20 days and mt-mKeima-positive cells were isolated by FACS. A clonally isolated cell line expressing suitable levels of the mt-mKeima reporter was selected for further experiments in this paper. This cell line is resistant to hygromycin (conferred by hTERT vector pGRN145), G418/geneticin/neomycin (conferred by the Cas9i vector HP138-neo) and blasticidin (conferred by the vector pAB01-pcDNA3.1-mt-mKeima-BlastR), and sensitive to puromycin.

### E3-targeting sgRNA library generation

sgRNAs targeting 606 Ubiquitin E3 ligases, as well as 243 non-targeting control (NTC) sequences, were designed by Azimuth2 (Doench *et al*, 2016) and the top four picks (on-target scores > 0.6) were chosen per gene, extended by 5 and 3 prime 3Cs homology and obtained as a pool from Twist Bioscience (Dataset EV1). The E3-targeting sgRNA library was made by 3Cs, as described previously (Wegner *et al*, 2019, 2020; Diehl *et al*, 2021). In brief, 3Cs-DNA was generated by mixing phosphorylated pre-annealed oligonucleotides (sgRNA-encoding) with ssDNA (library template plasmid). 3Cs-DNA was purified and desalted using the GeneJET Gel Extraction Kit (Thermo Fisher) and parts of it were analysed by gel electrophoresis alongside the dU-ssDNA template. The remaining 3Cs-DNA was electroporated into 10-beta electrocompetent *E. coli* (New England Biolabs) by using a Bio-Rad Gene Pulser and incubated overnight. The final sgRNA library plasmid DNA was purified using the Maxi Plasmid DNA Prep Kit (Qiagen).

### Virus production

To produce lentiviral particles for the screen, 70% confluent Lenti-X HEK293T (Clontech) cells were transfected in 10 cm$^2$ dishes with 16.5 μg E3 sgRNA library plasmid, 13.5 μg pPAX2 (12260; Addgene) and 5 μg pMD2G (12259; Addgene) using Lipofectamine 2000 according to the manufacturer's instructions. The media containing lentiviral particles were harvested 48 h later, centrifuged at 500 *g* for 5 min and small aliquots were snap-frozen in liquid nitrogen and stored at −80°C. The viral titre was determined by serial dilution and polybrene-mediated transduction of RPE1-Cas9i cells followed by selection in 5 μg/ml puromycin.

### CRISPR library screen

RPE1-Cas9i-mt-mKeima cells (75% confluency) were transduced on Day 0 with a lentiviral CRISPR library containing 2,667 unique sgRNA-encoding viruses, targeting 606 E3 ligases with 4 sgRNAs per gene plus 243 non-targeting controls (NTC) using polybrene (8 μg/ml). To achieve > 1,000× library coverage at a multiplicity of infection of 0.2, a total of 13.5 × 10$^6$ cells were transduced for each replicate and maintained throughout the screen. Twenty-four hours after transduction, puromycin selection (5 μg/ml) was initiated and maintained for 7 days. On day 3, Cas9 expression was induced with doxycycline (1 μg/ml) and maintained for 10 days. On day 14, "induced mitophagy" was initiated by treatment with antimycin A (1 μM) and oligomycin A (10 μM), while "basal mitophagy" was allowed to proceed in untreated cells. Approximately 24 h later, cells were harvested separately for FACS. A fraction of 2.7 × 10$^6$ cells was snap-frozen as a cell pellet in liquid nitrogen as an unsorted reference sample for both induced and basal mitophagy.

### Fluorescence-activated cell sorting and flow cytometry

For the CRISPR/Cas9 mitophagy screen, cells were harvested by trypsinization, washed, resuspended in FACS buffer (1% FBS/PBS) and stored on ice. Cells were sorted using a FACSAria III (BD Biosciences). Mitolysosomal versus mitochondrial mt-mKeima fluorescence was measured using dual-excitation ratiometric pH

measurements at 488 nm (pH 7) and 552 nm (pH 4) lasers with 695/40 nm and 610/20 nm emission filters, respectively. The top (high mitophagy) and bottom (low mitophagy) 20–30% of cells were collected in FACS buffer, washed with PBS and snap-frozen in liquid nitrogen. Four biologically independent experiments were sorted for each sample.

Samples for flow cytometry were prepared as above. Initial characterization of RPE1-Cas9i-mt-mKeima by flow cytometry was done using an LSR Fortessa (BD Biosciences) using 405 nm (pH 7) and 561 nm (pH 4) lasers with 610/20 nm emission filters. Data were analysed using Flowjo (v 10). High mitophagy gates were made by taking the top 10–15% of the control sample and applied to the other samples. Post-screen validation by flow cytometry was carried out using either a Bio-Rad ZE5 Cell analyser or a Sony MA900 cell sorter, using dual-excitation ratiometric pH measurements. pH 7 and pH 4 measurements were taken using 405 nm and 552 nm lasers, respectively, and 610/20 nm emission filters. Data were analysed using Flowjo (v 10). High mitophagy gates were made by taking the top 10–15% of the control sample and applying to the other samples. Post-screen validation was carried out at the Gene-Mill facility at the University of Liverpool.

### DNA extraction, library preparation and next-generation sequencing

Genomic DNA from sorted cells was purified using a Purelink Genomic DNA Kit (Invitrogen) and eluted in 50 μl nuclease-free water according to the manufacturer's instructions. Up to 5 μg genomic DNA was typically obtained from this process. Library amplification PCRs were performed using NEB Next Ultra II Q5 Master Mix (New England Biolabs) on all the extracted genomic DNA to attach Illumina sequencing adaptors and barcodes. A universal pool of eight forward primers were used for each sample along with a unique reverse primer containing a sample-specific barcode. The PCR cycling conditions used were 98°C for 30 s, 22 cycles of 98°C for 10 s, 63°C for 30 s and 65°C for 45 s, followed by a final extension of 65°C for 5 min. The library PCRs from two independent biological experiments were pooled for each of the two replicate samples submitted for NGS to ensure each NGS sample was derived from 2.4–5.4 × 10^6 cells, that is, a ~900–2,000× coverage of the E3 sgRNA plasmid library. Samples were purified using two rounds of Ampure-XP beads (Beckman Coulter) at a ratio of 0.8× according to the manufacturer's instructions. One microlitre purified product was run on a 3% agarose gel to ensure the primers were removed by purification and the amplicons were present at 375 bp. Samples were sequenced on a NovaSeq 6000 using Novaseq S4 PE150 chemistry (Novogene).

### Screen analysis

Raw sequencing reads were trimmed to 35 bp (containing unique sample barcodes) with BBTools using BBDuk (BBMap—Bushnell B.—sourceforge.net/projects/bbmap/) and untrimmed reads were removed. The ScreenProcessing pipeline (available at https://github.com/mhorlbeck/ScreenProcessing; Horlbeck et al, 2016) was used to count the number of uniquely mapped sgRNAs for each sample and to determine the enrichment of sgRNAs between samples. Briefly, the number of uniquely mapped sgRNAs for each

sample was counted, and sgRNAs with less than 50 reads were removed from the analysis. For each comparison, both a gene-level enrichment value ($Log_2$ fold change) calculated from the average of all four sgRNAs for each gene), as well as a Mann–Whitney *P*-value, reporting the probability that the distribution of enrichment values for all sgRNAs targeting a particular gene is different from the distribution of all 243 non-targeting control (NTC) gRNA values in the same replicate was generated (Bassik et al, 2013; Kampmann et al, 2013) (Dataset EV1). Plots were generated in R using ggplot2 and modified in Adobe Illustrator for style.

### Generation of FBXL4 KO cells

To generate FBXL4 control and knockout cell pools, RPE1-Cas9i-mt-mKeima cells were transduced with lentivirus-encoding FBXL4-targeting sgRNAs (sgRNA 1 (KO1)-TTGGTCAGAGAGACCTACGA or sgRNA 2 (KO2)-CTCAATGCAGAGGTAGTCCA) using polybrene (8 μg/ml) at a multiplicity of infection of 0.5. Twenty-four hours after transduction, cells were placed under puromycin (5 μg/ml) selection for 10 days. Half the pool was induced for Cas9 expression with doxycycline (1 μg/ml) for an additional 10 days. Genomic DNA was extracted from control and knockout pools, the sgRNA region was amplified using PCR and products were sent for sequencing. Knockout efficiency was determined using the ICE analysis from Synthego (Fig EV2C; https://ice.synthego.com/#/).

### Cell culture, transfection and siRNA interference

RPE1 cells were cultured in Dulbecco's modified Eagle medium DMEM/F12 (hTERT-RPE1) supplemented with 10% FBS and 1% non-essential amino acids, and Lenti-X HEK293T cells were cultured in Dulbecco's modified Eagle medium (DMEM (Gibco)) supplemented with 10% FBS. For siRNA experiments, cells were treated with 40 nM of non-targeting (NT1) or target-specific siRNA oligonucleotides (Dharmacon On-Target Plus), using Lipofectamine RNAi-MAX (Invitrogen, 13778030) according to manufacturer's instructions. The medium was exchanged after 6 h and cells were harvested 72 h after transfection. For plasmid transfections, Lipofectamine 2000 (Invitrogen, 11668019), Lipofectamine 3000 (Invitrogen, L3000001) or Genejuice (Merck Millipore, 70967) was used according to the manufacturers' protocol, unless otherwise stated. Transfection reactions were carried out for 16–24 h. For cycloheximide assays, cells were treated for the indicated time points with 100 μg/ml cycloheximide. Where indicated, 10 μM MG132 was added. Cell lines are regularly tested for mycoplasma contamination and cell stocks are authenticated.

### siRNA and plasmids

Sequences of siRNA used in this manuscript were as follows: FBXL4 (ON-TARGET Plus pool: 5′-GCAGUUGUGUCAUGAUUGA-3′, 5′-GGACAUAUUAGGAACAAGA-3′, 5′-GGAAUGGACAGUCUUAACA-3′, 5′-UUAGAAUUCUCGCUUGUUC-3′), VHL (siGENOME: 5′-CCGUAUGGCUCAACUUCGA-3′, 5′-AGGCAGGCGUCGAAGAGUA-3′, 5′-GCUCUACGAAGAUCUGGAA-3′, 5′-GGAGCGCAUUGCACAUCAA-3′), BNIP3 (ON-TARGET Plus pool: 5′-UCGCAGACACCACAAGAUA-3′, 5′-GAACUGCACUUCAGCAAUA-3′, 5′-GGAAAGAAGUUGAAAGCAU-3′, 5′-ACACGAGCGUCAUGAAGAA-3′), NIX (ON-TARGET Plus pool: 5′-GACC

AUAGCUCUCAGUCAG-3′, 5′-CAACAACAACUGCGAGGAA-3′, 5′-GAA
GGAAGUCGAGGCUUUG-3′, 5′-GAGAAUUGUUUCAGAGUUA-3′),
HIF1α (ON-TARGET Plus Pool: 5′-GAACAAAUACAUGGGAUUA-3′, 5′-
AGAAUGAAGUGUACCCUAA-3′, 5′-GAUGGAAGCACUAGACAAA-3′,
5′-CAAGUAGCCUCUUUGACAA-3′) and KDM2A (ON-TARGET Plus
pool: 5′-GAACAAUCCCAGCGGCAAA-3′, 5′-GUGUGCAAGACGUGGUA
UA-3′, 5′-CUGAGAAGAGAGACGCCAA-3′, 5′-CAAAGAGCUCCA
CGGGACA-3′). The plasmids pCDNA5 FRT TO FBXL4-3xFlag, pCDNA
FRT TO FBXL4 [R482W]-3xFlag, pCDNA5 FRT TO GFP-BNIP3 and
pCDNA5 FRT TO GFP-NIX were all purchased from Dundee MRC PPU.
The PAC sgRNA (5′-TGTCGAGCCCGACGCGCGTG-3′) was cloned into
the pSpCas9(BB)-2A-GFP (PX458; 48138; Addgene) vector while the
FBXL4 sgRNA-1 (5′-TTGGTCAGAGAGACCTACGA-3′) and FBXL4
sgRNA-2 (5′-TGGACTACCTCTGCATTGAG-3′) plasmids were cloned
into the pLenti-sgRNA (71409; Addgene) vector. All sgRNAs were
cloned using BsmBI restriction sites. The neomycin resistance gene
was removed from pcDNA3.1-mt-mKeima (Katayama *et al*, 2011) with
XmaI/BstBI (NEB) and the vector was treated with mung bean nucle-
ase and alkaline phosphatase calf intestinal (New England Biolabs).
The blasticidin resistance gene was isolated from a plasmid pKM808
using XhoI/SalI (New England Biolabs) and treated with a Quick
Blunting Kit (New England Biolabs) and the insert and vector were
ligated using Quick Ligase (New England Biolabs) to generate pAB01-
pcDNA3.1-mt-mKeima-BlastR.

### RNA isolation and qRT–PCR

Total RNA was isolated from RPE1 cells or RPE1 control and FBXL4
KO pools using a Qiagen RNA extraction kit (74106). cDNA was
generated from 1 μg RNA using RevertAir H Minus reverse tran-
scription (Thermo Scientific, 11541515) with RNasin (Promega,
N251S), oligo (dT) 15 primer (Promega, C1101) and PCR nucleotide
mix (Promega, U144B). Quantitative PCRs were performed in tripli-
cate using primers against BNIP3 (Pair 1; 5′-CCTCAGCATGAGG
AACACGA-3′; 5′-AAAAGGTGCTGGTGGAGGTT-3′; Pair 2; 5′-CCTT
CCATCTCTGCTGCTCTC-3′; 5′-TGGAGGTTGTCAGACGCCTT-3′), Nix
(Pair 1; 5′-AGGAAAATGAGCAGTCTCTGCC-3′; 5′-TGGAGGATG
AGGATGGTACG-3′; Pair 2; 5′-TGTCGTCCCACCTAGTCGAG-3′; 5′-
GCTGTTCATGGGTAGCTCCA-3′), Actin (5′-CACCTTCTACAATGAG
CTGCGTGTG-3′; 5′-ATAGCACAGCCTGGATAGCAACGTAC-3′), or
FBXL4 (Pair 1; 5′-TGAGATGTGTCCAAATCTACAGG-3′; 5′-GCTGAG
CAGTGCTGTTTGC-3′; Pair 2; 5′-AGCTGAAGTAAGATGGGAGAT-3′;
5′-ACTCGTATAAGATTTGTGGGGA-3′). Primer and cDNA reactions
were run with iTaq Mastermix (Bio-Rad, 172-5171) in a Biorad CFX
Connect real-time system. The mean cycle threshold (Ct) values
were normalized to actin (ΔCt = Ct target – Ct Actin), raised to the
exponent of $2^{-\Delta Ct}$ and normalized to the respective control cell line
to generate $2^{-\Delta\Delta Ct}$.

### Antibodies and reagents

Antibodies and other reagents are as follows: anti-actin (66009,
1:10,000, Proteintech), anti-BNIP3 (ab109362 1:1,000 WB, 1:100 IF,
Abcam), anti-NIX (12396, 1:1,000 WB, 1:250 IF, Cell signalling tech-
nology), anti-HIF1α (NB100-134, 1:1,000, Novus Bio techne), anti-
VHL (68547, 1:1,000, Cell signalling technology), anti-TOMM20
(612278, 1:500 IF, BD transduction), anti-TOMM40 (NBP2-38289,
1:1,000, Novus Bio techne), anti-Cathepsin D (219361, 1:2,000, Cal

Biochem), anti-Flag (F3165, 1:1,000 WB; 1:250 IF, Sigma-Aldrich),
anti-Flag (F7425, 1:1,000, Sigma-Aldrich), anti-Flag M2 agarose
affinity gel (A2220, Sigma-Aldrich), anti-GFP (In house, 1:1,000),
anti-phospho-ubiquitin (Ser65) (62802, 1:1,000, Cell Signalling tech-
nology), anti-PINK1 (6946, 1:1,000, Cell signalling technology),
anti-LC3 (5F10, 1:200, Nanotools), anti-p62 (610833, 1:1,000, BD
Transduction), anti-CUL1 (71-8700, 1:1,000, Invitrogen), MLN4924
(C-1231, Chemgood), MG132 (Sigma-Aldrich), cycloheximide
(Sigma-Aldrich), oligomycin A (75351; Sigma-Aldrich) and anti-
mycin A (A8674; Sigma-Aldrich).

### Preparation of cell lysates and western blot analysis

Cultured cells were lysed with RIPA buffer (150 mM NaCl, 1%
sodium deoxycholate, 10 mM Tris–Cl pH 7.5, 0.1% SDS and 1% Tri-
ton X-100) supplemented with MPI (mammalian protease inhibitor)
cocktail (P8340; Sigma-Aldrich) and Phosstop (04906837001;
Roche). Proteins were resolved using SDS–PAGE (Invitrogen
NuPage gel 4–12%), transferred to nitrocellulose membrane
(10600002; Amersham), stained with Ponceau S staining solution
(P7170; Sigma-Aldrich), blocked in 5% milk (Marvel) or 5% BSA
(41-10-410; First Link) in TBS (20 mM Tris–Cl, pH 7.6 and 150 mM
NaCl) supplemented with Tween-20 (10485733; Thermo Fisher Sci-
entific) and probed with primary antibodies overnight. Visualization
and quantification of Western blots were performed using IRdye
800CW- (goat 926-32214, mouse 926-32212 and rabbit 926-32213),
and 680LT- (goat 926-68024, mouse 926-68022 and rabbit 926-
68023) coupled secondary antibodies and an Odyssey infrared scan-
ner (LI-COR Biosciences). For western blot quantitation, raw signal
values were obtained from image studio software following back-
ground subtraction. Raw values for each condition were normalized
to the sum of the quantitated raw values from each individual blot,
and statistical analysis was conducted in PRISM 9.

### Co-immunoprecipitation

hTERT-RPE1 FlpIN TREX cells were transfected with FBXL4-Flag WT
or R482W and GFP-tagged BNIP3 or NIX using Lipofectamine 3000,
and expression was induced with 0.1 μg/ml doxycycline for 16 h prior
to lysis in NP40 buffer (0.5% NP40, 25 mM Tris pH 7.5, 100 mM
NaCl and 50 mM NaF) supplemented with MPI, Phosstop and CAA
(2-chloroacetamide, 50 mM). Lysates (250–800 μg protein) were then
incubated with 10 μl of either GFP nano-trap beads or Flag affinity
gel. For GFP nano-trap pulldowns, incubation was for 1 h before
beads were washed in IP wash buffer (0.1% NP40, 25 mM Tris pH
7.5, NaCl 100 mM and NaF 50 mM). Bound proteins were eluted in
sample buffer (62.5 mM Tris pH 6.8, 3% SDS, 10% glycerol and 3.2%
β-mercaptoethanol). For Flag pulldowns, incubation was for 2 h
before beads were washed with IP wash buffer (0.1% NP40, 25 mM
Tris pH 7.5, NaCl 100 mM and NaF 50 mM). Bound proteins were
eluted in Flag peptide (Sigma, F4799, 150 ng/μl) for 30 min. Eluted
proteins were diluted in sample buffer and analysed by western blot.

### TUBE pulldowns

For TUBES pulldowns, cells were lysed in TUBES lysis buffer
(50 mM Tris–HCl pH7.5, 150 mM NaCl, 1 mM EDTA, 1% NP40 and
10% glycerol) supplemented with MPI, Phosstop and 10 mM NEM.

Lysates (300–500 μg protein) were then incubated with 10–20 μl of TUBES or control resin overnight. Beads were washed with 0.1% TBST and bound proteins were eluted in sample buffer (62.5 mM Tris pH 6.8, 3% SDS, 10% glycerol and 3.2% β-mercaptoethanol).

**Immunofluorescence and live-cell imaging**

Cells were fixed in 4% paraformaldehyde in PBS and permeabilized with 0.2% Triton X-100 in PBS. Coverslips were blocked in goat serum, and then cells were stained with primary antibodies. Proteins were visualized using AlexaFluor-488- or AlexaFluor-647-coupled secondary antibodies. Fixed coverslips were imaged using either a Zeiss LSM800 or LSM900 Airyscan (63× oil, acquisition software Zen blue). Images were processed using Adobe Photoshop (v 22.1.1) and Fiji (v 2.1.0) software. All images were acquired sequentially. For live-cell imaging of mt-mKeima, cells were seeded onto IBIDI μ-Dishes (81156; IBIDI) 2 days prior to imaging with a 3i Marianas spinning-disk confocal microscope (40× or 63× oil objective, NA 1.4, Photometrics Evolve EMCCD camera, acquisition software Slide Book 3i v3.0). Images were acquired sequentially (445 nm excitation, 617/73 nm emission; 561 nm excitation, 617/73 nm emission). Analysis of basal mitophagy levels was performed using the "mitoQC Counter" plug-in in Fiji (v 2.1.0) software, as previously described (Montava-Garriga *et al*, 2020), using the following parameters: radius for smoothing images = 1.25, ratio threshold = 0.8 and red channel threshold = mean + 1 SD. Mitophagy analysis was performed for two to three independent experiments with minimum of 47 cells per condition.

**Statistical analysis**

Bar graphs indicate mean and standard deviation. Statistical significance was determined with an unpaired *t*-test (Figs 7G, EV2A and EV3C) or one-way ANOVA with either Dunnett's (Figs 2B, 5B, 6D and H, and EV2B) or Tukey's (Figs 3B, E and F, and 6G and I) multiple-comparisons tests using GraphPad Prism 9. *P*-values are represented as *$P < 0.05$, **$P < 0.01$, ***$P < 0.001$ and ****$P < 0.0001$.

# Data availability

This study includes no data deposited in external repositories.

**Expanded View** for this article is available online.

## Acknowledgements

We thank Jonathon Pines (ICR London) and Iain Cheeseman (MIT) for generously providing hTERT-RPE1-FlpIN and CT33-hTERT-RPE1-Cas9i cell lines, respectively, for this project. We also thank Francesco Barone for help in evaluating and selecting the single-cell m-mKeima expressing cell clone, Jin-Rui (Amos) Liang for help and advice with the screen processing pipeline and Robert Taylor (Newcastle, UK) for useful discussions. We are grateful to the Liverpool University Centre for Cell Imaging, the Liverpool Cell Sorting and Isolation Facility and the Liverpool GeneMill for access to instrumentation. Initial observations of MLN4924-induced mitophagy were made by Elena Marcassa and Jane Jardine (University of Nantes, France) while working in the Liverpool laboratory. This work was supported by a Parkinson's UK Grant (G-1902) and the Deutsche Forschungsgemeinschaft (DFG, German Research Foundation; Project ID 259130777–SFB 1177). KRM is supported by a studentship from the Medical Research Council (MRC) Discovery Medicine North (DiMeN) Doctoral Training Partnership (MR/N013840/1). MJC is the recipient of a Royal Society Industry Fellowship (INF\R2\212031). AJB is the recipient of funding from the European Union's Horizon 2020 research and innovation programme under the Marie Sklodowska-Curie grant agreement No 897783.

## Author contributions

**Hannah Elcocks:** Conceptualization; data curation; formal analysis; investigation; visualization; methodology; writing – original draft; writing – review and editing. **Ailbhe J Brazel:** Conceptualization; data curation; formal analysis; investigation; visualization; methodology; writing – original draft; writing – review and editing. **Katy R McCarron:** Formal analysis; investigation; visualization; methodology; writing – original draft. **Manuel Kaulich:** Resources; methodology. **Koraljka Husnjak:** Resources; methodology. **Heather Mortiboys:** Conceptualization; supervision; funding acquisition; project administration. **Michael J Clague:** Conceptualization; formal analysis; supervision; funding acquisition; investigation; writing – original draft; project administration; writing – review and editing. **Sylvie Urbé:** Conceptualization; data curation; formal analysis; supervision; funding acquisition; investigation; visualization; methodology; writing – original draft; project administration; writing – review and editing.

## Disclosure and competing interests statement

MK is a co-founder, shareholder and chief officer of Vivlion GmbH. MJC and SU are academic founders and SAB members of ENTACT Bio.

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
