## [Review Process File · The EMBO Journal]

FBXL4 ubiquitin ligase deficiency promotes mitophagy by elevating NIX levels

Hannah Elcocks, Ailbhe Brazel, Katy McCarron, Manuel Kaulich, Koraljka Husnjak, Heather Mortiboys, Michael Clague, and Sylvie Urbe

DOI: [10.15252/emboj.2022112799](https://doi.org/10.15252/emboj.2022112799)

Corresponding author(s): Sylvie Urbe (urbe@liv.ac.uk) , Michael Clague (clague@liv.ac.uk)

Review Timeline:

Submission Date:	11th Oct 22
Editorial Decision:	24th Nov 22
Revision Received:	17th Mar 23
Accepted:	9th Apr 23

Editor: Hartmut Vodermaier

Transaction Report:

Prof. Sylvie Urbe
University of Liverpool
Molecular and Cellular Physiology
University of Liverpool
Crown Street
Liverpool, Merseyside L69 3BX
United Kingdom

24th Nov 2022

Re: EMBOJ-2022-112799
FBXL4 deficiency promotes mitophagy by elevating NIX

Dear Sylvie,

Thank you again for submitting your work on FBXL4 in basal mitophagy suppression to The EMBO Journal. With some delay caused by the referees having to simultaneously review two co-submitted manuscripts, I have now received the below-copied three reports. As you will see, the referees consider this work timely and potentially interesting, but also note various aspects that would still require strengthening, especially in order for the study to be able to stand on its own. Should you be able to satisfactorily address these concerns, we would be happy to pursue a revised version of the study further.

In light of the overlapping findings by the co-submitted study, I realize that it may not be necessary to respond to every single point of criticism experimentally; but in order to better define which experiments would be crucial and which would lie beyond the scope of the present manuscript, it would be helpful to briefly discuss this with you via email or video call. To facilitate such discussion, I would invite you to send me a tentative point-by-point response and revision plan, once you will have had a chance to consider the reports together with your colleagues.

I should add that we could also offer extension of the default three-months revision period if needed, and that competing/overlapping work appearing here or elsewhere in the meantime will have no negative impact on our final decision on your study.

Detailed information on preparing, formatting and uploading a revised manuscript can be found below and in our Guide to Authors. Thank you again for the opportunity to consider this work for The EMBO Journal, and I look forward to hearing from you in due time.

With kind regards,

Hartmut

9) Digital image enhancement is acceptable practice, as long as it accurately represents the original data and conforms to community standards. If a figure has been subjected to significant electronic manipulation, this must be clearly noted in the figure legend and/or the 'Materials and Methods' section. The editors reserve the right to request original versions of figures and the original images that were used to assemble the figure. Finally, we generally encourage uploading of numerical as well as gel/blot image source data; for details see: embopress.org/page/journal/14602075/authorguide#sourcedata

At EMBO Press, we ask authors to provide source data for the main manuscript figures. Our source data coordinator will contact you to discuss which figure panels we would need source data for and will also provide you with helpful tips on how to upload and organize the files.

In the interest of ensuring the conceptual advance provided by the work, we recommend submitting a revision within 3 months (22nd Feb 2023). Please discuss the revision progress ahead of this time with the editor if you require more time to complete the revisions. Use the link below to submit your revision:

Link Not Available

Referee #1:

This a review of the manuscript by Elcocks et al., however, another manuscript by different authors was also co-submitted and reviewed. Although both manuscripts will be reviewed separately, the rationale and significance are the same for both. The manuscripts show that FBXL4 regulates mitophagy via turnover of the mitochondrial membrane proteins NIX and BNIP3. Both NIX and BNIP3 are well known regulators of mitophagy, and FBXL4 has also been shown to negatively regulate this process, yet the link between FBXL4 mitophagy and BNIP3/NIX had not been previously made. This discovery is important for two reasons as BNIP3/NIX-dependent mitophagy was previously thought to be regulated largely by transcriptional upregulation but also that fact that FBXL1 is mutated in a mitochondrial-related disease and the discovery of a BNIP3/NIX involvement suggests a potential therapeutic pathway. Therefore, these works are exciting and publishable.

Elcocks et al., discover FBXL4 as a negative regulator of "basal" mitophagy via a CRSIPR E3 ligase screen and then go onto validate this hit as a regulator of NIX-mediated mitophagy. On the whole, the experiments are clear and justify the conclusion that FBXL4 regulates NIX levels and in turn mitophagy. Additionally, the identification of MLN compound as a strong pharmacological inducer of NIX-mitophagy will be a useful reagent for the field.

It is very important for the field that independent groups show similar data, which is the case here. However, I do note that the

other manuscript is more substantial in nature and has carried out many of the experiments suggested here. Regardless, reviewing this as a stand-alone manuscript, I feel some more data is required and as the authors now have the tools to go into a greater depth on the mechanism, I have the following suggestions.

Main points

- 1) Some more information on how FBXL4 is regulating NIX/BNIP3 levels would be highly beneficial. Can the authors rescue the KO mitophagy phenotype by stably re-expressing WT and mutant forms of FBXL4 (e.g. truncation mutants that no longer interact with NIX/BNIP3 as well as disease-associated mutations)? How does the R482W mutation work in these kinds of experiments?
- 2) Similar to above, can the authors rescue the FBXL4 mitophagy phenotype by rescuing NIX KOs with NIX mutations that don't bind FBXL4, or that can no longer become ubiquitinated?
- 3) Is the mitophagy phenotype upon loss of FBXL4 solely due to increased NIX levels? For example, can overexpression of NIX in WT cells phenocopy the KO cells with respect to mitophagy?
- 4) One of the authors main rationales for carrying out the screen was to identify E3 ligases that were sensitive to USP30 inhibition, yet no further USP30 involvement with the hits was analysed. Is it possible that USP30 could be regulating basal mitophagy via deubiquitination of NIX? For example, does overexpression of USP30 alter NIX levels and can this be rescued by USP30 inhibitors?

Minor points

- 5) Some of the explanations in the text are a little brief and could be expanded. For example, the description for Figure 5 is very limited and the authors do not explain the FBXL4 R482W mutation sufficiently.
- 6) Are the authors able to detect endogenous FBXL4 with an antibody? If not, perhaps this should be stated. How was KO of FBXL4 confirmed - this should be by genomic sequencing at the least.
- 7) Statistical analyses are missing for Figure 2.
- 8) The Western blot data in EV2D would benefit from quantitation.
- 9) Please quantify the mitophagy data in Figure 6A.
- 10) I think the authors are overstating the findings of the manuscript in the abstract. Perhaps the following statement could be toned down: "Our study enables a full understanding of the aetiology of early onset mitochondrial encephalomyopathy that is supported by analysis of a disease associated mutation"?

Referee #2:

In this manuscript, Elcocks et al. report VHL and FBXL4, two components of the Cullin-RING E3 ubiquitin ligase (CRL) complexes, as negative regulators of mitochondria-specific autophagy (mitophagy) under basal conditions. In mammalian cells, mitophagy is mediated via two distinct mechanisms recruiting the autophagy machinery to mitochondria: one relying on ubiquitin chains conjugated to mitochondrial surface proteins and autophagy adaptors, and the other relying on specific mitochondria-anchored receptors for autophagic proteins. The former includes Parkin/PINK1-mediated mitophagy that degrades depolarized mitochondria. Although recent studies reveal that mitophagy can occur in Parkin knockout mice and fruit flies under steady-state conditions, molecular mechanisms regulating Parkin-independent basal mitochondrial clearance remains largely unknown. Here, the authors performed a CRISPR/Cas9 screen for ubiquitin E3 ligase components regulating mitophagy in the absence of Parkin and found that mitophagy was enhanced in cells lacking VHL and FBXL4. The VHL-containing CRL has previously been suggested to promote degradation of HIF-1/2 alpha, two transcription factors that drive expression of the mitophagy receptors NIX and BNIP3. In this study, the authors focused on FBXL4 and demonstrated that the protein levels of NIX and BNIP3 were increased in cells depleting FBXL4, whereas their mRNA levels and the HIF-1 alpha protein level were hardly affected by loss of FBXL4. These results suggest that FBXL4 acts in suppression of NIX/BNIP3-mediated mitophagy in a manner independent of VHL. The authors also found that FBXL4 interacts with NIX and BNIP3, and that overexpression of FBXL4, but not FBXL4(R482W), a variant associated with mitochondrial DNA depletion syndrome 13, caused a decrease in expression of NIX and BNIP3. Despite the fact that loss of FBXL4 leads to upregulation of NIX and BNIP3, depletion of NIX, but not BNIP3, canceled mitophagy acceleration in FBXL4 knockout cells. Finally, the authors established the CRL inhibitor MLN4924 as an inducer of NIX-mediated mitophagy.

The results in this study are solid with proper quantification and could provide new insights into NIX-mediated mitophagy under basal conditions. However, additional data are needed to verify that NIX and BNIP3 are direct substrates of the FBXL4-dependent ubiquitination (ideally by in vitro assays and mass spectrometry). In addition, whether acceleration of NIX/BNIP3-mediated basal mitophagy is the primary cause of mitochondrial dysfunction in cells expressing FBXL4(R482W) has not been explored. In conclusion, this study would significantly be strengthened if the authors clarify these issues and address the following points.

Specific points:

1. In addition to Figure 4A and 4B, the authors should perform western blot analysis to confirm if NIX and BNIP3 are indeed ubiquitinated in a manner dependent on FBXL4.
2. Related to Figure 4C-F, the authors should investigate whether mitophagy is enhanced in cells expressing FBXL4(R482W),

which can be canceled by loss of NIX/BNIP3.

3. The authors should test if FBXL4(R482W) is defective in formation of the CRL complex, or substrate recognition for NIX and BNIP3.

Minor point:

1. In Figure 5A, the authors should add representative microscopic images of cells undergoing mitophagy.

Referee #3:

The manuscript by Elcocks et al describes novel regulation mechanism of BNIP3L/NIX- and BNIP3- receptor mediated mitophagy. The authors have performed Crispr/Cas9 screen of E3 ligases and found FBXL4 to be a negative regulator of selective receptor-mediated mitophagy. They have demonstrated that upon FBXL4 depletion both BNIP3L/NIX and BNIP3 are accumulating in the cells leading to increased mitophagy induction. This is further analysed to show that lack of ubiquitination of BNIP3L/NIX leads to accumulation of the receptor thus promoting mitophagy. In addition, it is shown that MLN4924 global Cullin RING ligase inhibitor is a strong inducer of receptor mediated mitophagy.

Although the study is very interesting and would significantly improve the understanding of the regulation of receptor-mediated mitophagy, I find it not yet suitable for publication in the presented form.

Here are specific comments:

- 1) The authors produced either FBXL4 knockout (KO) or knockdown (KD) cells but did not show KO/KD was successful. I find it essential to show that protein is indeed not being produced in these cells therefore, anti-FBXL4 staining needed.
- 2) The authors use mtKeima to monitor mitophagy that has been widely used, but LC3 accumulation at mitochondria should be added as independent method to screen receptor-mediated mitophagy.
- 3) Could authors show that accumulation of BNIP3 and NIX is indeed happening on the mitochondrial level in additional method beside immunofluorescence staining? Perhaps, mitochondrial subcellular fractionation could be performed.
- 4) Furthermore, the manuscript would be additionally improved if not just the initiation of mitophagy, but the mitophagy flux (progression) would be monitored.
- 5) In Fig 4 the authors are using GFP-BNIP3L/NIX and -BNIP3 constructs. It can be seen that the bands are not sharp but very broad? What is the reason for this? Other publications using GFP-tagged receptors do not report problem with constructs? Also it looks like less protein could be used in pull downs (especially BNIP3), blots look overloaded.
- 6) In Fig 4A there is double band of Flag-FBXL4 in input but not pull-down. This is not the case in 4B. This should be commented. Also, for easier interpretation of the data in figures 4A i 4B it would be great if blots are organized in the same order (e.g. FBXL4 blot up, receptors down).
- 7) Minor comment: BNIP3-only lane in Fig 4A is missing.
- 8) Figure 4C-F: labeling of the quantification graphs should be simplified to easier monitor which bar corresponds to which ImF image. Also, in 4D, R482W panel merged image is not the representative image of the individual stainings (similar comment goes for Fig 6A).
- 9) In Fig 4D, NIX panel looks equally low expressed in ctrl as in FBXL4-WT panel.
- 10) The authors comment that NIX is more stable than BNIP3 (Fig 7) but this is not seen on the presented images (NIX error bars are just bigger and wb of NIX more intense).
- 11) The neddylation of FBXL4 is discussed in results but no proof on neddylation has been shown.

Additional comments:

- Have authors examined LIR mutants/LIR phosphorylation mutants or even TM mutants?
- Recent work by Wilhelm et al, 2022, EMBO journal, have shown that BNIP3L/NIX is needed for pexophagy (similar treatments were used) - it would be great addition to check if FBXL4 influences BNIP3L/NIX- mediated pexophagy in the similar manner.

EMBOJ-2022-112799R

FBXL4 deficiency promotes mitophagy by elevating NIX

Point-by-point response in reference to the editorial letter and referee's comments of 24th Nov 2022

We thank the editor for his guidance and all three reviewers for their insightful comments, which undoubtedly helped us improve our manuscript. Whilst we have given serious consideration to each point made, our prioritisation of these points reflects the consensus arising from a structured online discussion with the editor.

Referee #1:

This a review of the manuscript by Elcocks et al., however, another manuscript by different authors was also co-submitted and reviewed. Although both manuscripts will be reviewed separately, the rationale and significance are the same for both. The manuscripts show that FBXL4 regulates mitophagy via turnover of the mitochondrial membrane proteins NIX and BNIP3. Both NIX and BNIP3 are well known regulators of mitophagy, and FBXL4 has also been shown to negatively regulate this process, yet the link between FBXL4 mitophagy and BNIP3/NIX had not been previously made. This discovery is important for two reasons as BNIP3/NIX-dependent mitophagy was previously thought to be regulated largely by transcriptional upregulation but also that fact that FBXL1 is mutated in a mitochondrial-related disease and the discovery of a BNIP3/NIX involvement suggests a potential therapeutic pathway. Therefore, these works are **exciting and publishable**.

Elcocks et al., discover FBXL4 as a negative regulator of "basal" mitophagy via a CRSIPR E3 ligase screen and then go onto validate this hit as a regulator of NIX-mediated mitophagy. On the whole, the experiments are clear and **justify the conclusion that FBXL4 regulates NIX levels and in turn mitophagy**. Additionally, the identification of MLN compound as a **strong pharmacological inducer of NIX-mitophagy will be a useful reagent** for the field.

It is very important for the field that independent groups show similar data, which is the case here. However, I do note that the other manuscript is more substantial in nature and has carried out many of the experiments suggested here. Regardless, reviewing this as a stand-alone

manuscript, I feel some more data is required and as the authors now have the tools to go into a greater depth on the mechanism, I have the following suggestions.

We thank the reviewer for these supportive comments. We have also been gratified by the congruent results reported by others. We think our work is differentiated from the parallel study cited by the referee by arriving at FBXL4 in an unbiased manner through a CRISPR-based screen of E3 ligases regulating mitophagy. This in itself is a very substantial piece of work and the first such screen of mitophagy under basal conditions and in Parkin-deficient cells. Our results also allow us to highlight the dual regulation of NIX by VHL and FBXL4 via different mechanisms.

Main points

- 1) Some more information on how FBXL4 is regulating NIX/BNIP3 levels would be highly beneficial.

We appreciate this consideration, and are pleased to provide additional data supporting the notion that FBXL4 regulates particularly NIX protein stability by targeting it for ubiquitin dependent proteasomal degradation. We show that 1) NIX (and BNIP3) transcript levels are not affected by deletion of FBXL4 (**Fig 3G**); 2) NIX (and BNIP3) protein turnover is increased in FBXL4 KO cells (**Fig 7B-D**, and **EV3E-G**) and 3) NIX ubiquitylation is strongly reduced in FBXL4 KO cells (**New Fig 7F-G**). Unfortunately, in our experimental system, BNIP3 ubiquitylation is much harder to detect and is less sensitive to proteasome inhibition, which will warrant further examination.

Can the authors rescue the KO mitophagy phenotype by stably re-expressing WT and mutant forms of FBXL4 (e.g. truncation mutants that no longer interact with NIX/BNIP3 as well as disease-associated mutations)? How does the R482W mutation work in these kinds of experiments?

We agree that this would be a welcome addition to our manuscript, however certain practicalities of our system make this challenging. Our knockout cells stably express mt-mKeima, which has to be imaged live. The GFP-tagged FBXL4 constructs do not target accurately to mitochondria in distinction to FBXL4-Flag. Thus we cannot simply conduct a rescue experiment with transiently expressed FBXL4/mutant but would have to generate another set of stable cell lines. Our mitokeima FBXL4 KO cell lines are already resistant to four antibiotics (Hyg, Bast, Neo, Puro), making this a trickier task than usual. We have attempted this experiment in multiple configuration (using frankenbody co-expression to detect Flag-FBXL4 in live cells; transducing transiently with lentivirus expressing GFP from an IRES), but have not been able to achieve sufficient viable co-expressing cells to address this point in our system.

We clearly show that NIX and BNIP3 levels are restored by (Flag-tagged) WT FBXL4 but not the R428W mutant (**Fig 4F-H**). We also show that the increased levels of mitophagy in our mt-Keima expressing FBXL4 KO cells are dependent on elevated NIX levels; we now provide representative images as well as highly quantitative flow cytometry data (**New Fig 5A and Fig 5B**). Thus we strongly feel that we have addressed the key point at hand.

2) Similar to above, can the authors rescue the FBXL4 mitophagy phenotype by rescuing NIX KOs with NIX mutations that don't bind FBXL4, or that can no longer become ubiquitinated?

We have not used NIX knockout cells in our experiments, rather we have shown that depleting NIX can rescue the FBXL4 KO mitophagy phenotype. It is unclear what an FBXL4-binding deficient NIX mutant would reveal in this context as there would be no FBXL4 present to regulate NIX.

3) Is the mitophagy phenotype upon loss of FBXL4 solely due to increased NIX levels? For example, can overexpression of NIX in WT cells phenocopy the KO cells with respect to mitophagy?

There is an extensive existing literature which shows that elevated NIX promotes mitophagy - we do not think that repeating such experiments will generate new insight. We have shown in **Figure 5** that knocking down NIX restores mitophagy in FBXL4 KO cells back to basal levels, which would indicate that upregulation of NIX is the major mechanism by which the loss of FBXL4 promotes mitophagy.

4) One of the authors main rationales for carrying out the screen was to identify E3 ligases that were sensitive to USP30 inhibition, yet no further USP30 involvement with the hits was analysed.

It is true that our prior work, including studies of USP30 demonstrating that ubiquitin dependent mitophagy remains relevant in cells that do not express Parkin, has focused our attention on the role of other E3s with respect to mitophagy. However in this screen we have taken no steps to identify USP30 sensitive factors, thus the statement regarding the rationale for our screen is not accurate, and **we have amended the text** to avoid this misconception (**Page 3, second paragraph**). We intend to pursue positive E3 regulators we have identified in future studies, but have nevertheless adopted a policy of openness in revealing the full screen results.

Is it possible that USP30 could be regulating basal mitophagy via deubiquitination of NIX? For example, does overexpression of USP30 alter NIX levels and can this be rescued by USP30 inhibitors?

This is certainly an interesting idea. USP30 depletion up-regulates basal mitophagy in Parkin deficient cells, however we have shown in our previous manuscripts that this was strictly dependent on PINK1. We have also recently shown that USP30 inhibition with a selective compound (Cmpd39) does not upregulate basal NIX or BNIP3 levels Barone et al. Life Sci. Alliance, 2023. Overexpression of a DUB confined to a membrane compartment will inevitably influence ubiquitylation- but more often through a mass action effect than any inherent specificity. A thorough investigation of a potential role for USP30 in the regulation of NIX lies outside the scope of this manuscript.

Minor points

5) Some of the explanations in the text are a little brief and could be expanded. For example, the description for Figure 5 is very limited and the authors do not explain the FBXL4 R482W mutation sufficiently.

We appreciate this point and agree that our description was a little too brief at this point. In response to Reviewer 2 (Minor point 1) we have provided additional data for figure 5 in form of representative images to accompany the highly quantitative flow cytometry data. We have **expanded the text accompanying this figure and have also expanded the description** of the FBXL4 R428W mutation in the text and figure legends (**Page 7, third paragraph** and **Page 8 first paragraph** respectively).

6) Are the authors able to detect endogenous FBXL4 with an antibody? If not, perhaps this should be stated. How was KO of FBXL4 confirmed - this should be by genomic sequencing at the least.

Unfortunately neither we, nor our colleagues in Australia, have been able to detect endogenous FBXL4 with commercially available antibodies. **We have now remarked on this in the text and have also added qRT-PCR data in New Figure EV2A (Page 6, Paragraph 3)**. The knockout status of the cell pools was confirmed by sequencing and this information should have been included in the manuscript. We now have corrected this oversight (Experimental procedures and **New Fig EV2C**).

7) Statistical analyses are missing for Figure 2.

We are now providing statistics for **Figure 2B**, as well as for other western blots, where appropriate.

8) The Western blot data in EV2D would benefit from quantitation.

We now have added a quantitation to this figure (**New Fig EV2E**).

9) Please quantify the mitophagy data in Figure 6A.

We would like to point out that the same conditions shown in Figure 6A are also incorporated in **Figure 6H** in a flow cytometry based mitophagy experiment that captures a much larger number of cells than we can analyse by imaging. Both the flow cytometry analysis in 6H (quantitatively) and the representative images in 6A (qualitatively) show that the increase of mitophagy achieved with MLN4924 is very large and comparable to that observed with AO.

10) I think the authors are overstating the findings of the manuscript in the abstract. Perhaps the following statement could be toned down: "Our study enables a full understanding of the aetiology of early onset mitochondrial encephalomyopathy that is supported by analysis of a disease associated mutation"?

Agreed - we have revised this statement. This now reads: "Our study contributes to an understanding of the aetiology of early onset mitochondrial encephalomyopathy that is supported by analysis of a disease associated mutation". We have also attenuated similar statements elsewhere in the text (Intro Page 4, Discussion Page 10).

Referee #2:

In this manuscript, Elcocks et al. report VHL and FBXL4, two components of the Cullin-RING E3 ubiquitin ligase (CRL) complexes, as negative regulators of mitochondria-specific autophagy (mitophagy) under basal conditions. In mammalian cells, mitophagy is mediated via two distinct mechanisms recruiting the autophagy machinery to mitochondria: one relying on ubiquitin chains conjugated to mitochondrial surface proteins and autophagy adaptors, and the other relying on specific mitochondria-anchored receptors for autophagic proteins. The former includes Parkin/PINK1-mediated mitophagy that degrades depolarized mitochondria. Although recent studies reveal that mitophagy can occur in Parkin knockout mice and fruit flies under steady-state conditions, molecular mechanisms regulating Parkin-independent basal mitochondrial clearance remains largely unknown. Here, the authors performed a CRISPR/Cas9 screen for ubiquitin E3 ligase components regulating mitophagy in the absence of Parkin and found that mitophagy was enhanced in cells lacking VHL and FBXL4. The VHL-containing CRL has previously been suggested to promote degradation of HIF-1/2 α , two transcription factors that drive

expression of the mitophagy receptors NIX and BNIP₃. In this study, the authors focused on FBXL₄ and demonstrated that the protein levels of NIX and BNIP₃ were increased in cells depleting FBXL₄, whereas their mRNA levels and the HIF-1 alpha protein level were hardly affected by loss of FBXL₄. These results suggest that FBXL₄ acts in suppression of NIX/BNIP₃-mediated mitophagy in a manner independent of VHL. The authors also found that FBXL₄ interacts with NIX and BNIP₃, and that overexpression of FBXL₄, but not FBXL₄(R482W), a variant associated with mitochondrial DNA depletion syndrome 13, caused a decrease in expression of NIX and BNIP₃. Despite the fact that loss of FBXL₄ leads to upregulation of NIX and BNIP₃, depletion of NIX, but not BNIP₃, canceled mitophagy acceleration in FBXL₄ knockout cells. Finally, the authors established the CRL inhibitor MLN4924 as an inducer of NIX-mediated mitophagy.

The results in this study are solid with proper quantification and could provide new insights into NIX-mediated mitophagy under basal conditions. However, additional data are needed to verify that NIX and BNIP₃ are direct substrates of the FBXL₄-dependent ubiquitination (ideally by *in vitro* assays and mass spectrometry).

We now show that NIX ubiquitylation is strongly reduced in FBXL₄ KO cell pools, further strengthening our interpretation that NIX is a direct substrate of FBXL₄ (**New Fig 7F and G**). In our experimental system, BNIP₃ ubiquitylation is much harder to detect and is less sensitive to proteasome inhibition, which will warrant further examination.

In vitro assays, reconstituting CRL activities on specific substrates are not straight forward, and even in laboratories with the required expertise can take years to establish. This is even more true in this particular, rare, case, where both the FBOX protein and its substrate are integral membrane proteins. Without doubt, the relationship between FBXL₄ and BNIP₃ and NIX deserves a detailed structure-function analysis, which lies however outside our expertise and the scope of this manuscript.

Mass spectrometry cannot provide evidence that something is a direct substrate. In principle it could detect a change in ubiquitylation status but who is to say that this a direct effect. Of course the most parsimonious interpretation would be that it is - but that is also true of our experiments (deletion of FBXL₄ enhances NIX turnover, which can be rescued by re-expression of FBXL₄ and not a mutant thereof).

In addition, whether acceleration of NIX/BNIP₃-mediated basal mitophagy is the primary cause of mitochondrial dysfunction in cells expressing FBXL₄(R482W) has not been explored.

Depletion of mitochondria is the major, well-characterised signature associated with FBXL4 mutation. We think the evidence is at least as strong as that linking mitophagy defects to Parkinson's disease in Parkin mutant patients. Given the practical limitations of our cell system, we have not generated FBXL4 (R482W) knockin cell lines to address this point, and a full analysis of the mitochondrial functionality in the context of this mutant would go beyond the scope of this manuscript. While our paper was under review, another manuscript has been uploaded on BioRxiv that provides further evidence in the context of FBXL4 deficient mice (+/- NIX/BNIP3 KO) that the upregulation of NIX and BNIP3 are the underlying cause of the associated disease aetiology (bioRxiv 2022.11.11.516094). We believe that all three studies are in excellent agreement on this point, whilst arriving at it from different starting points.

In conclusion, this study would significantly be strengthened if the authors clarify these issues and address the following points.

Specific points:

1. In addition to Figure 4A and 4B, the authors should perform western blot analysis to confirm if NIX and BNIP3 are indeed ubiquitinated in a manner dependent on FBXL4.

We now show that NIX ubiquitylation is strongly reduced in FBXL4 KO cell pools, further strengthening our interpretation that NIX is a direct substrate of FBXL4 (**New Fig 7F and G**). In our experimental system, BNIP3 ubiquitylation is much harder to detect and is less sensitive to proteasome inhibition, which will warrant further examination.

2. Related to Figure 4C-F, the authors should investigate whether mitophagy is enhanced in cells expressing FBXL4(R482W), which can be canceled by loss of NIX/BNIP3.

See response to Reviewer 1, Major point 1.

3. The authors should test if FBXL4(R482W) is defective in formation of the CRL complex, or substrate recognition for NIX and BNIP3.

Our experiments have indicated that against our expectation, the interaction between FBXL4 and its substrate NIX is only marginally affected (**Fig 4C**). This is intriguing since this, as well as most other disease associated mutations, are located at the substrate interacting end (the LRR) of this CRL adapter. Instead, we find that it is the interaction with the core CRL component SKP1 that is clearly affected by this mutation in our pulldown experiments (**Figure 4D, E**). Under our experimental conditions, we do not detect Cul1 in these pulldowns. We are mindful of the fact that pulldown experiments can be misleading when it comes to small changes in affinity that

nevertheless can be functionally important in a physiological context. We have reflected this in our interpretation in the text and suggest that future structure function and detailed biochemical analyses elsewhere will be required to fully answer this question (Page 9).

Minor point:

1. In Figure 5A, the authors should add representative microscopic images of cells undergoing mitophagy.

As requested by the reviewer, we now provide representative microscopic images of cells (New Fig 5A) to go alongside the flow cytometry based experiment for which we provide quantitation and stats (New Fig 5B).

Referee #3:

The manuscript by Elcocks et al describes novel regulation mechanism of BNIP₃L/NIX- and BNIP₃- receptor mediated mitophagy. The authors have performed Crispr/Cas9 screen of E₃ ligases and found FBXL₄ to be a negative regulator of selective receptor-mediated mitophagy. They have demonstrated that upon FBXL₄ depletion both BNIP₃L/NIX and BNIP₃ are accumulating in the cells leading to increased mitophagy induction. This is further analysed to show that lack of ubiquitination of BNIP₃L/NIX leads to accumulation of the receptor thus promoting mitophagy. In addition, it is shown that MLN4924 global Cullin RING ligase inhibitor is a strong inducer of receptor mediated mitophagy.

Although the study is very interesting and would significantly improve the understanding of the regulation of receptor-mediated mitophagy, I find it not yet suitable for publication in the presented form.

Here are specific comments:

1) The authors produced either FBXL₄ knockout (KO) or knockdown (KD) cells but did not show KO/KD was successful. I find it essential to show that protein is indeed not being produced in these cells therefore, anti-FBXL₄ staining needed.

As explained above in response to reviewer 2, the currently available antibodies against FBXL₄ do not detect the endogenous protein. **We now remark on this in the text (Page 6, Paragraph 3).** We also have included the sequencing information showing that the KO cells have indeed been edited and we have provided qRTPCR data (New Figure EV2A and C) to demonstrate that FBXL₄ mRNA is down regulated in the FBXL₄ knockdown cells.

2) The authors use mtKeima to monitor mitophagy that has been widely used, but LC₃ accumulation at mitochondria should be added as independent method to screen receptor-mediated mitophagy.

We are focusing here on basal mitophagy, which monitors low stochastic mitophagy events. LC₃ accumulation on mitochondria is usually seen in cells over expressing Parkin upon acute induction of mitophagy using a depolarising trigger, and even then only weakly. In our experience most of the LC₃ puncta following acute mitochondrial depolarisation are recruited to non-mitochondrial compartments which have presumably been damaged by ROS. As the reviewer points out, the mitophagy reporter we use has been widely employed to provide a more specific picture of selective mitophagy as LC₃ levels in cells can fluctuate in response to secondary stress which complicates the analysis. In addition, LC₃ colocalisation with mitochondria does not discriminate between LC₃ actually recruited to mitochondria and LC₃ located on ER-mitochondrial junctions. Thus in our opinion, this is not an appropriate nor necessary experiment to validate the fact that we are visualising mitophagy.

3) Could authors show that accumulation of BNIP₃ and NIX is indeed happening on the mitochondrial level in additional method beside immunofluorescence staining? Perhaps, mitochondrial subcellular fractionation could be performed.

We believe the immunofluorescence microscopy we provide is unequivocal on this point (**Figures 3C and 4C,D and EV3B**). Subcellular fractionation can be done but since the mitochondria fractions are not pure, we fail to see in what way this will enhance our manuscript.

4) Furthermore, the manuscript would be additionally improved if not just the initiation of mitophagy, but the mitophagy flux (progression) would be monitored.

The mitokeima reporter reports on the acidic environment of the mitolysosome. Traditional flux experiments compare the readout in the presence and absence of a vacuolar proton pump inhibitor which will abolish the acidic environment and nullify the readout of this reporter.

5) In Fig 4 the authors are using GFP-BNIP₃L/NIX and -BNIP₃ constructs. It can be seen that the bands are not sharp but very broad? What is the reason for this? Other publications using GFP-tagged receptors do not report problem with constructs? Also it looks like less protein could be used in pull downs (especially BNIP₃), blots look overloaded.

We are also intrigued by the banding pattern of our GFP-NIX and BNIP3 constructs. We have provided additional pulldown experiments that reveal some degradation products that may contribute to some of the broad appearance of the banding pattern (**Fig 4C**). We note that BNIP3 and NIX are known to dimerise and are post-translationally modified, both of which could contribute to their behaviour on SDS-PAGE. We are planning to continue our work on these proteins in the future and hope to gain further insight in their regulation.

6) In Fig 4A there is double band of Flag-FBXL4 in input but not pull-down. This is not the case in 4B. This should be commented. Also, for easier interpretation of the data in figures 4A i 4B it would be great if blots are organized in the same order (e.g. FBXL4 blot up, receptors down).

We have noticed the double band in the input and it is indeed intriguing that this higher molecular weight pool of FBXL4 does not appear to co-immunoprecipitate with NIX or BNIP3 (**Fig 4A**, also seen now in new **Fig 4C**) whilst it is immunoprecipitated with the Flag antibody. As requested, we now also comment on this in the text and in the Figure legends.

We prefer the current arrangement of our western blots in A and B, which are in each case showing the protein that is immunoprecipitated (bait) below the one that is queried (prey).

7) Minor comment: BNIP3-only lane in Fig 4A is missing.

That is correct. The key control for the IP component of this experiment is to show that FBXL4 is not simply pulled down by the GFP vector alone. We have also shown a condition just expressing GFP-NIX without FBXL4 to demonstrate that the banding we see in the Flag western blot of the IP derives from from FBXL4-Flag as it is only seen in the samples that express FBXL4-Flag.

8) Figure 4C-F: labeling of the quantification graphs should be simplified to easier monitor which bar corresponds to which ImF image.

Agreed, our labelling was confusing - we have changed the labelling of those graphs (New **Fig 4F, G**).

Also, in 4D, R482W panel merged image is not the representative image of the individual stainings (similar comment goes for Fig 6A).

This is incorrect - in each case the merged images are merges of the individual panels (Old Fig 4D is now **New Fig 4G**).

9) In Fig 4D, NIX panel looks equally low expressed in ctrl as in FBXL4-WT panel.

Yes we completely agree. The experiment shows that the FBXL₄ wild-type protein is able to reset NIX levels back to baseline levels. (Old Fig 4D is now **New Fig 4G**).

10) The authors comment that NIX is more stable than BNIP₃ (Fig 7) but this is not seen on the presented images (NIX error bars are just bigger and wb of NIX more intense).

It is a bit unclear here which exact panel the reviewer comments on. The rate of loss of NIX versus BNIP₃ in response to CHX in the Control cells is following a steeper trajectory for BNIP₃ than for NIX (**Fig 7B-D**). We have re-visited the quantitation in **Figures 7C, D and EV3F, G**, and whilst some large error bars remain, each experiment reveals the same trend. We have also revised the text to reflect the fact that we do see an effect also on BNIP₃ stability in the FBXL₄ KO cells (**Page 9, first paragraph**).

11) The neddylation of FBXL₄ is discussed in results but no proof on neddylation has been shown.

FBXL₄ is not itself neddylated but we have now included a representative Western Blot of CUL₁, showing the well established inhibition of its neddylation by MLN4924 (**Fig 6B**).

Additional comments:

- Have authors examined LIR mutants/LIR phosphorylation mutants or even TM mutants?

No we have not, although we do agree that this warrants further investigation.

- Recent work by Wilhelm et al, 2022, EMBO journal, have shown that BNIP_{3L}/NIX is needed for pexophagy (similar treatments were used) - it would be great addition to check if FBXL₄ influences BNIP_{3L}/NIX- mediated pexophagy in the similar manner.

We agree that this is an interesting question but we believe this would distract from the focus of this manuscript which concerns mitophagy.

Prof. Sylvie Urbe
University of Liverpool
Molecular and Cellular Physiology
University of Liverpool
Crown Street
Liverpool, Merseyside L69 3BX
United Kingdom

9th Apr 2023

Re: EMBOJ-2022-112799R
FBXL4 ubiquitin ligase deficiency promotes mitophagy by elevating NIX adaptor levels

Dear Sylvie,

Thank you for submitting your final revised manuscript for our consideration. I am pleased to inform you that in light of the below-copied positive re-reviews by two of the original referees, we have now accepted it for publication in The EMBO Journal.

Your article will be processed for publication in The EMBO Journal by EMBO Press and Wiley, who will contact you with further information regarding production/publication procedures and license requirements. You will also be provided with page proofs after copy-editing and typesetting of main manuscript and expanded view figure files.

Should you be planning a Press Release on your article, please get in contact with embojournal@wiley.com as early as possible, in order to coordinate publication and release dates.

Yours sincerely,

Hartmut

Referee #1:

This is a re-review of the manuscript by Elcocks et al., concerning the role of FBXL4 in regulating mitophagy via BNIP3/NIX stabilisation. The extra work and clarifications are welcome, though it is a shame that the authors were unable to carry out some of the deeper mechanistic experiments suggested. However the data from this work, combined with that from the other two manuscripts, makes for compelling evidence that FBXL4 is regulating mitophagy via BNIP3/NIX levels. Given that one manuscript is already out, I do not feel the need to delay this one further by requesting additional work.

Referee #2:

In this revised manuscript, Elcocks et al. provided additional data and descriptions to clarify most of the points suggested by the referees. Specifically, the authors performed pull-down assays and demonstrated that the mitophagy-promoting factor NIX is ubiquitinated in a manner dependent on FBXL4, a component of the Cullin-RING E3 ubiquitin ligase (CRL) complex (Fig. 7F and G), and that FBXL4(R482W), a variant associated with mitochondrial DNA depletion syndrome 13, is capable of NIX binding, but partially defective in its interaction with Skp1, a component of the CRL complex (Fig. 4C-E). Whether FBXL4-mediated regulation of NIX-driven mitophagy is physiologically relevant in vivo remains to be addressed, however, this study will ensure a positive impact on a wide spectrum of biomedical studies related to cellular proteostasis and mitochondrial quality control.